# BIDCell: Biologically-informed self-supervised learning for segmentation of subcellular spatial transcriptomics data

Xiaohang Fu [1,2,3,4,5,8], Yingxin Lin [1,3,4,5,8], David M. Lin[6], Daniel Mechtersheimer[1,3,4], Chuhan Wang[2,3,5], Farhan Ameen [1,3,4], Shila Ghazanfar [1,3,4], Ellis Patrick [1,3,4,5,7], Jinman Kim[2,3,5] & Jean Y. H. Yang [1,3,4,5] ✉

Recent advances in subcellular imaging transcriptomics platforms have enabled high-resolution spatial mapping of gene expression, while also introducing significant analytical challenges in accurately identifying cells and assigning transcripts. Existing methods grapple with cell segmentation, frequently leading to fragmented cells or oversized cells that capture contaminated expression. To this end, we present BIDCell, a self-supervised deep learning-based framework with biologically-informed loss functions that learn relationships between spatially resolved gene expression and cell morphology. BIDCell incorporates cell-type data, including single-cell transcriptomics data from public repositories, with cell morphology information. Using a comprehensive evaluation framework consisting of metrics in five complementary categories for cell segmentation performance, we demonstrate that BIDCell outperforms other state-of-the-art methods according to many metrics across a variety of tissue types and technology platforms. Our findings underscore the potential of BIDCell to significantly enhance single-cell spatial expression analyses, enabling great potential in biological discovery.

High-throughput spatial omics technologies are at the forefront of modern molecular biology, and promise to provide topographic context to the wealth of available transcriptomic data. Recent breakthroughs in profiling technology have revolutionised our understanding of multicellular biological systems, and the collection of Subcellular Spatial Transcriptomics (SST) technologies (e.g. 10x Genomics Xenium[1]; NanoString CosMx[2]; BGI Stereo-seq[3]; and Vizgen MERSCOPE) now offer the promise to tackle biological problems that were previously inaccessible and better understand intercellular communication by preserving tissue architecture. Depending on the commercial platforms, these ultra-high resolution, spatially resolved single-cell data contain mixtures of nuclear, cytoplasmic, and/or cell membrane signals, and create new data challenges in information extraction. More specifically, the aim is to ensure all available data can be capitalised to automatically and accurately distinguish the boundaries of individual cells, as the fundamental goal of SST technologies is to understand how single-cell transcriptomes behave in situ within a given tissue[4].

Limited attempts have been made to address these data challenges and to date, three conceptual categories have emerged. The first employs morphological operations originally designed for lower-resolution imaging technologies such as microscopy. Within this

[1]School of Mathematics and Statistics, The University of Sydney, Sydney, NSW 2006, Australia. [2]School of Computer Science, The University of Sydney, Sydney, NSW 2006, Australia. [3]Sydney Precision Data Science Centre, University of Sydney, Sydney, NSW 2006, Australia. [4]Charles Perkins Centre, The University of Sydney, Sydney, NSW 2006, Australia. [5]Laboratory of Data Discovery for Health Limited (D24H), Science Park, Hong Kong SAR, China. [6]Department of Biomedical Sciences, Cornell University, Ithaca, NY 14850, USA. [7]The Westmead Institute for Medical Research, Sydney, NSW 2145, Australia. [8]These authors contributed equally: Xiaohang Fu, Yingxin Lin. ✉e-mail: jean.yang@sydney.edu.au

category, initial nuclei segmentation is accomplished with a nuclear marker, using thresholding or pretrained models such as Cellpose[5] and Mesmer[6]. Cell boundaries are then identified using either morphological expansion by a prespecified distance[1] or using a watershed algorithm on a mask of the cell bodies[3]. Chen et al. applied a global threshold to the density of all molecules in SST data to estimate the cell body mask. The limitation of Cellpose[5] and similar approaches is that they were primarily designed for microscopy modalities and fluorescent markers, so they may not always be suitable for SST due to dissimilar visual characteristics.

Secondly, an alternative approach to cell segmentation does not identify cell boundaries directly, but classifies or clusters individual transcripts into distinct measurement categories that pertain to cells. These include segmentation-free and transcript-based methods, as exemplified by Baysor[7], StereoCell[8], pciSeq[9], Sparcle[10], and ClusterMap[11]. However, a key limitation of these approaches is their assumption that expression of all RNAs within a cell body are homogeneous, and in the case of Baysor, that cell shapes (morphologies) can be well approximated with a multivariate normal prior. This can result in visually unrealistic segmentations that do not correspond well to imaging data.

Thirdly, more recent approaches have begun to leverage deep learning (DL) methods. DL models such as U-Net[12] have provided solutions for many image analysis challenges. However, they require ground truth to be generated for training. DL-based methods for SST cell segmentation include GeneSegNet[13] and SCS[14], though supervision is still required in the form of initial cell labels or based on hard-coded rules. Further limitations of existing methods encountered during our benchmarking, such as lengthy code runtimes, are included in Supplementary Table 1. The self-supervised learning (SSL) paradigm can provide a solution to overcome the requirement of annotations. While SSL-based methods have shown promise for other imaging modalities[15,16], direct application to SST images remains challenging. SST data are considerably different from other cellular imaging modalities and natural images (e.g., regular RGB images), as they typically contain hundreds of channels, and there is a lack of clear visual cues that indicate cell boundaries. This creates new challenges such as (i) accurately delineating cohesive masks for cells in densely-packed regions, (ii) handling high sparsity within gene channels, and (iii) addressing the lack of contrast for cell instances.

While these morphological and DL-based approaches have shown promise, they have not fully exploited the high-dimensional expression

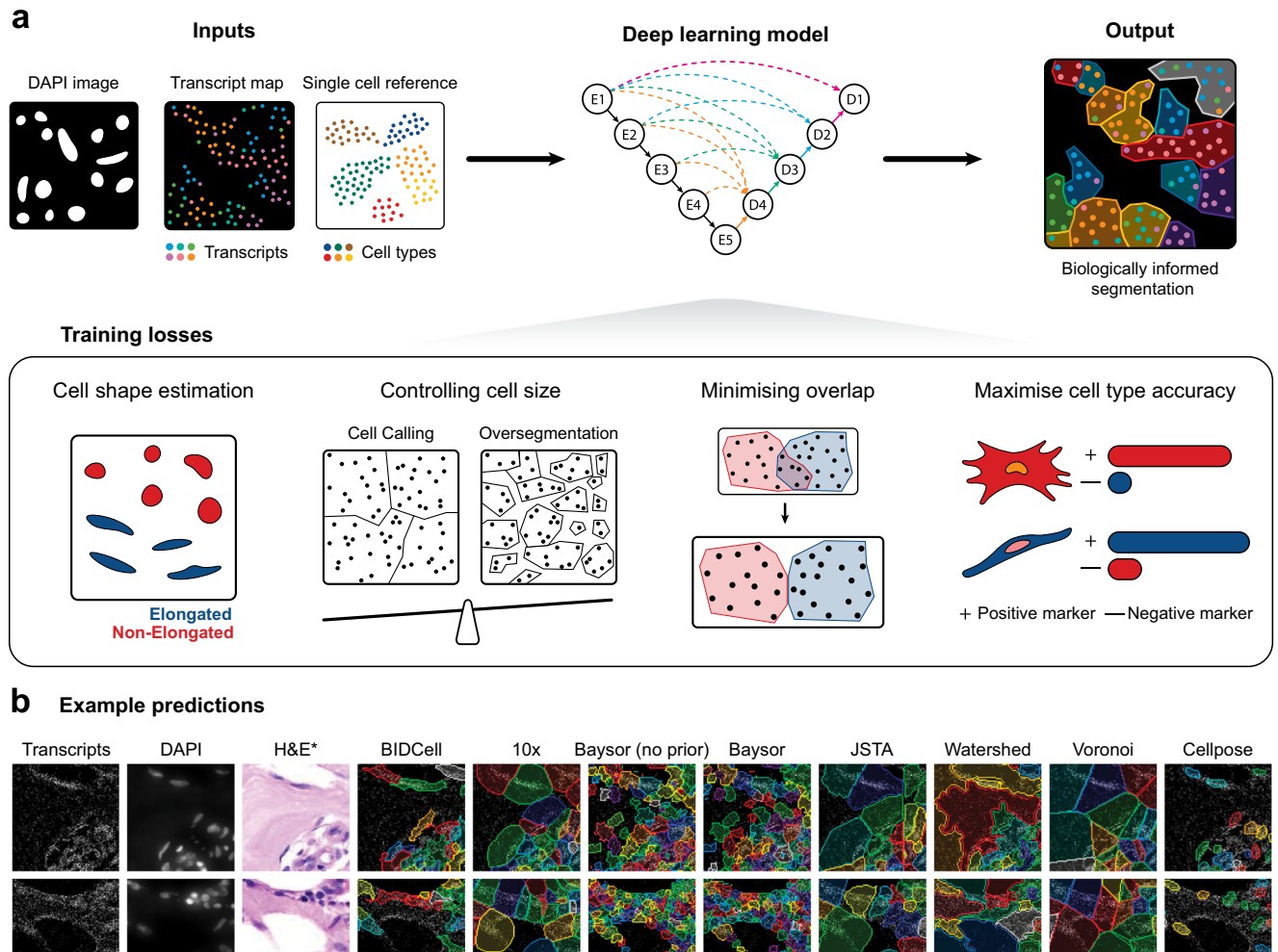

**Fig. 1 | BIDCell framework. a** Schematic illustration of the BIDCell framework and the loss functions used for training. In the deep learning model, E1 to E5 and D1 to D4 are respectively the encoding and decoding layers, while the connectivity between layers to each decoding layer is indicated by arrows of a unique colour (e.g., green for D3). **b** Comparative illustration of the predictions from BIDCell and other cell segmentation methods on the public Xenium-BreastCancer1 dataset. BIDCell captures cell morphologies with better correspondence to the input images, with a more diverse set of cell shapes that include elongated types. The H&E images are provided for illustration purposes only and were not used as an input for any of the methods shown.

information contained within SST data. It has become increasingly clear that relying solely on imaging information may not be sufficient to accurately segment cells. There is growing interest in leveraging large, well-annotated scRNA-seq datasets[17], as exemplified by JSTA[18], which proposed a joint cell segmentation and cell type annotation strategy. While much of the literature has emphasised the importance of accounting for biological information such as transcriptional composition, cell type, and cell morphology, the impact of incorporating such information into segmentation approaches remains to be fully understood.

Here, we present a biologically-informed deep learning-based cell segmentation (BIDCell) framework (Fig. 1a), that addresses the challenges of cell body segmentation in SST images through key innovations in the framework and learning strategies. We introduce (a) biologically-informed loss functions with multiple synergistic components; and (b) explicitly incorporate prior knowledge from single-cell sequencing data to enable the estimation of different cell shapes. The combination of our losses and use of existing scRNA-seq data in supplement to subcellular imaging data improves performance, and BIDCell is generalisable across different SST platforms. Along with the development of our segmentation method, we created a comprehensive evaluation framework for cell segmentation, CellSPA, that assesses five complementary categories of criteria for identifying the optimal segmentation strategies. This framework aims to promote the adoption of new segmentation methods for novel biotechnological data.

## Results

### BIDCell: Incorporating biological insights using deep learning to improve cell shape representation

BIDCell is a DL-based cell segmentation method that identifies each individual cell and all its pixels as a cohesive mask. BIDCell uses subcellular spatial transcriptomic maps, corresponding DAPI images, and relevant average expression profiles of cell types from single-cell sequencing datasets; the latter is obtained from public repositories such as the Human Cell Atlas. Given the lack of ground truth and visual features that indicate cell boundaries in the SST images, BIDCell instead focuses on the relationships between the high-dimensional spatial gene expressions and cell morphology. The BIDCell framework automatically derives supervisory signals from the input data and/or predicted segmentations, which is an approach to learning that we borrow from SSL.

To achieve this, we designed multiple loss functions that represent various criteria based on biological knowledge, that work synergistically to produce accurate segmentations (Fig. 1a; see Methods and Supplementary Materials for a detailed description). BIDCell learns to use the locations of highly- and lowly-expressed marker genes to calibrate the segmentation to capture higher "cell expression purity", thereby ensuring transcripts within each cell share the same profile. Furthermore, BIDCell captures local expression patterns using a data-driven, cell-type-informed morphology. We found that the eccentricity measure of nuclei could reveal diverse cell morphologies that correspond to established knowledge, such as elongated morphologies for fibroblasts (Supplementary Fig. 1). By capturing a diverse set of cell shapes and leveraging marker information from previous single-cell experiments (Table 1), BIDCell generates superior segmentations (Fig. 1b and Supplementary Figs. 2 and 3), and overcomes the limitations of many existing methods (Table 2) that rely primarily on SST image intensity values for cell segmentation.

We further ensure the integrity of cell segmentations by proposing three other cooperative loss functions. Appropriate cell sizes are supported by capturing expression patterns local to nuclei using guidance from cell-type informed morphologies (cell-calling), while ensuring the cohesiveness of cell instances (over-segmentation) and enhancing segmentation in densely-populated regions (overlap loss).

BIDCell also leverages expression patterns within nuclei to guide the identification of cell body pixels.

We investigated removing individual losses in an ablation study with Xenium-BreastCancer1 data (Supplementary Figs. 4, 5). Our investigation shows that the losses work synergistically; e.g., there was a marked increase in purity F1 relative to the amount of captured transcripts when the losses were combined. With the inclusion of single-cell data (which informs the positive and negative losses, and contributes to the ability to predict elongated cell shapes), performance improved considerably, particularly in purity metrics and correlation to Chromium data. The use of single-cell data helped the model to better capture transcripts that are more biologically meaningful within cells. By default, the weights of the losses are all 1.0 and do not need to be tuned for BIDCell to perform well, though further fine-tuning is possible (Supplementary Fig. 6). The popular UNet 3+[19] serves as the segmentation backbone architecture in BIDCell, though this is not a requirement and it may be replaced with alternative architectures (Supplementary Fig. 7).

### CellSPA comprehensive evaluation framework captures diverse sets of metrics of segmentation aspects across five complementary categories

To ensure an unbiased comparison, we introduce a Cell Segmentation Performance Assessment (CellSPA) framework (Fig. 2a) that captures cell segmentation metrics across five complementary categories. These categories, detailed in Fig. 2a and Supplementary Table 2, include (i) baseline characteristics at both the cell and gene levels; (ii) measures of segmented cell expression, where we assess the "expression purity" of our assigned segmented cells based on how well transcripts within the segmented cell share a similar expression profile; (iii) measures of baseline cell characteristics in its spatial environment, including spatial region diversity and corresponding diversity in morphology; (iv) a measure of contamination between nearest neighbours (Supplementary Fig. 8); and (v) measures of replicability.

Using CellSPA, we compared the performance of BIDCell with several recently developed methods for the segmentation of SST data. These methods included classical segmentation-based approaches such as simple dilation, watershed, and Voroni; and transcript-based approaches including Baysor. Additionally, we evaluated JSTA[18], which attempts to jointly determine cell (sub) types and cell segmentation based on an extension from the traditional watershed approach. In all comparisons, we limited the computational time to within 72 h, which we deemed a practical requirement for the solutions provided by each approach (see Discussion).

To ensure the minimal appropriateness of segmented cells, we examine a series of quality control (QC) statistics. As an illustrative example using Xenium-BreastCancer1 data, we segmented cells using BIDCell, generating 100,000 number of cells, with 53.4% of transcripts assigned (Fig. 2b). We first confirm that the total number of transcripts per cell and the number of genes per cell were greater in the whole cell (cell body + nuclei) compared to just the nuclei (Fig. 2c and Supplementary Fig. 9).

Similarly, using the percentage of cells expressing each gene between the nuclei vs. the cell body, we further evaluate the level of information presented in the nuclei and the cell body from the gene level (Fig. 2d). We find that the segmented cells of some of the methods (e.g. Baysor) did not yield any additional transcript information beyond that of the nuclei, where we see a tight concordance (lying on a 45-degree line) between the segmented cell body and the cell nuclei. However, BIDCell, 10x, Cellpose, and JSTA are all able to capture additional transcript information. Moving forward, we will focus on methods that provide "additional" information to the nuclei, with an emphasis on the ability to better capture cell boundaries.

Lastly, we examine the cell morphology of the segmented cells against the segmented nuclei, including cell area, elongation, compactness, sphericity, convexity, eccentricity, solidity and circularity

**Table 1 | Single-cell RNA-seq references used in this study**

| Data collection | Data | # of cell types | Source |
|---|---|---|---|
| TISCH-BRCA | GSE110686 | 17 | http://tisch.comp-genomics.org/gallery/?cancer=BRCA&species=Human |
| | GSE114727_10X | | |
| | GSE114727_inDrop | | |
| | GSE138536 | | |
| | GSE143423 | | |
| | GSE176078 | | |
| | SRP114962 | | |
| | EMTAB8107 | | |
| | GSE148673 | | |
| | GSE150660 | | |
| Chromium-BreastCancer | Single Cell Gene Expression Flex (FRP) | 22 | https://www.10xgenomics.com/products/xenium-in-situ/preview-dataset-human-breast |
| Mouse brain | Allen brain map | 59 | https://portal.brain-map.org/atlases-and-data/rnaseq/mouse-whole-cortex-and-hippocampus-smart-seq |
| HLCA | Banovich_Kropski_2020 | 50 | https://beta.fastgenomics.org/p/hlca |
| | Krasnow_2020 | | |
| | Lafyatis_Rojas_2019 | | |
| | Meyer_2019 | | |
| | Misharin_2021 | | |
| | Misharin_Budinger_2018 | | |
| | Teichmann_Meyer_2019 | | |
| TISCH-NSCLC | EMTAB6149 | 1 | http://tisch.comp-genomics.org/gallery/?cancer=SCLC&species=Human |
| | GSE117570 | | |
| | GSE127465 | | |
| | GSE143423 | | |
| | GSE148071 | | |
| | GSE150660 | | |
| SKCM atlas | GSE115978 | 15 | http://tisch.comp-genomics.org/gallery/?cancer=SKCM&species=Human |
| | GSE120575 | | |
| | GSE123139 | | |
| | GSE139249 | | |
| | GSE148190 | | |
| | GSE72056 | | |
| | GSE134388 | | |
| | GSE159251 | | |
| | GSE166181 | | |
| | GSE179373 | | |

**Table 2 | Summary of existing methods used for comparison**

| Types | Method | Nuclei segmentation | Cell body segmetation | Public code | Reference |
|---|---|---|---|---|---|
| Nuclei | 10x (Nuclei) | 10x | NA | N/A | |
| | Cellpose (Nuclei) | Cellpose | NA | Version 2.1.1 | 5 |
| Adapted from classical approach | Cellpose nuclei dilated | Cellpose | Dilation | OpenCV (v4.6.0) | |
| | Voronoi | Cellpose | Voronoi expansion | SciPy library (v1.9.3) | |
| | Watershed | Cellpose | Watershed algorithm | OpenCV (v4.6.0) | |
| Deep learning-based | 10x | 10x | 10x | N/A | |
| | BIDCell | Cellpose | BIDCell | Version 4494e02 | |
| | Cellpose cell | Cellpose | Cellpose | Version 2.1.1 | 5 |
| | JSTA | Cellpose | JSTA | Version ccce064 | 18 |
| Transcript-based | Baysor | N/A or Cellpose | Baysor | Version 0.5.2 | 7 |

(See Methods and Supplementary Fig. 10). Through these metrics, we are able to identify the outliers of the segmented cells, such as cells with extremely large areas in JSTA, Voronoi and Watershed in the sparse areas (Supplementary Fig. 11). We illustrate that as intended from our cell-mask, BIDCell has cell morphology that is highly correlated with the nuclei morphology (Fig. 2e). Furthermore, we find that segmented cells from BIDCell exhibit more diverse cell morphology characteristics compared to other methods (Supplementary Fig. 12).

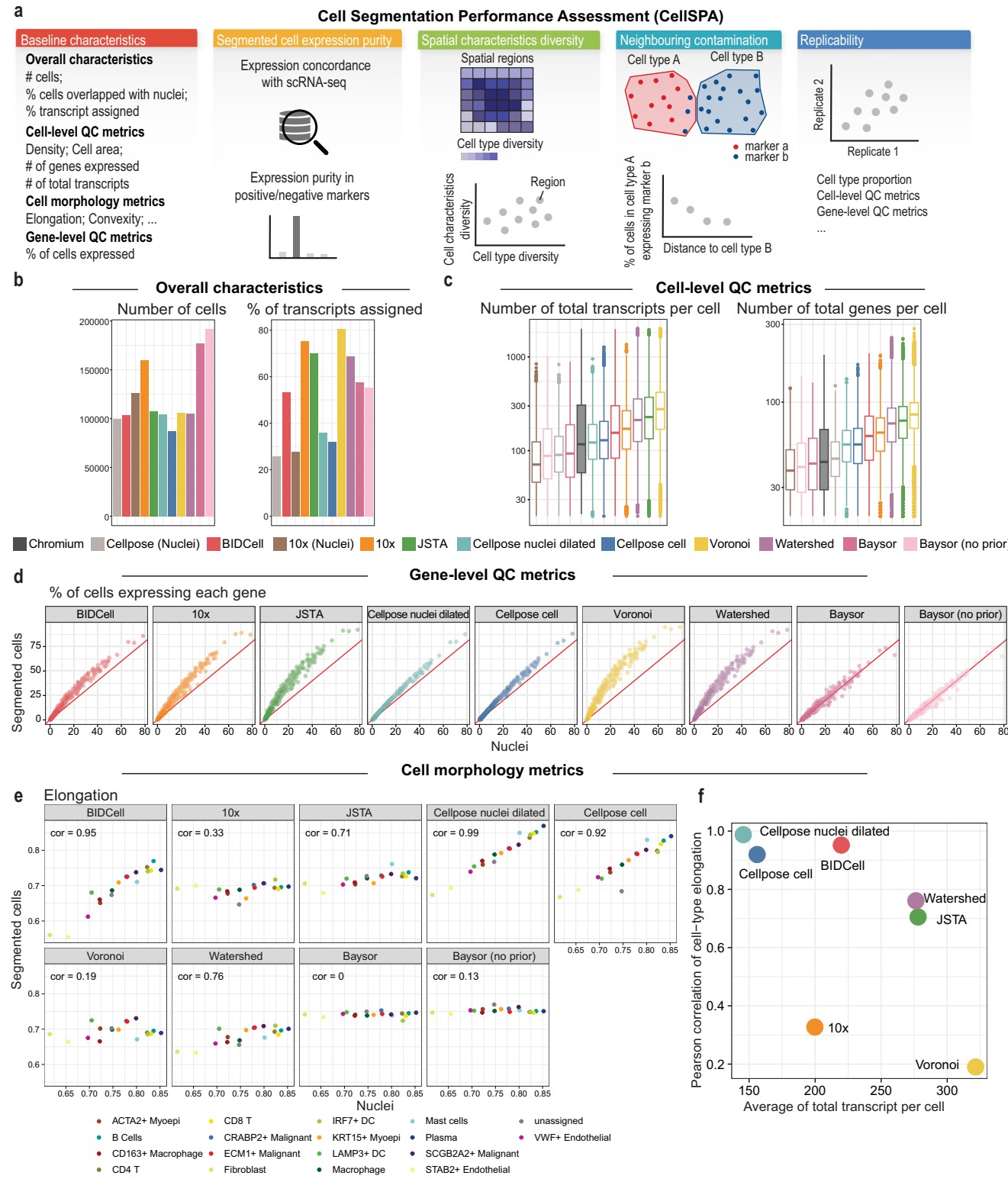

**BIDCell captures improved purity of cell expression, leading to less contamination from neighbouring cells**

To determine whether various cell segmentation methods can improve spatial resolution without sacrificing detection efficiency, we first compare the correlation between cell type signatures in the Xenium and Chromium V2 platforms for Xenium-BreastCancer1 data (Fig. 3a). We observed that the performance of correlation for average expression between the spatial and sequencing profile ranges between 0.72 and 0.8 across all methods. Interestingly, we observe a trade-off

between the size of the cell (average total transcript per cell) and the level of correlation. Fig. 3a demonstrates the importance of employing two metrics to quantify segmentation performance. While Cellpose achieved the highest Pearson correlation overall, BIDCell achieved the highest Pearson correlation among methods that detect a similar number of transcripts as Chromium data (i.e., cell sizes that are more similar to segmentation of the cell body as opposed to the nuclei). Similar results are shown in the average percentage of expressed genes (Fig. 3b). Furthermore, Fig. 3c highlights a high level of consistency in

**Fig. 2 | CellSPA performance evaluation framework. a** Schematic showing the cell segmentation evaluation framework with five complementary categories. **b** Bar plots showing overall characteristics, including the number of cells [left], and the number of transcripts [right] for each of the 11 methods. **c** Boxplots of cell-level quality metrics with total number of transcripts [left] and total number of genes [right]. The number points for each box includes the number of cells detected by each method (N = Chromium: 22,294; Cellpose (nuclei): 99,693; BIDCell: 103,209; 10x (nuclei): 126,515; 10x: 160,254; JSTA: 107,131; Cellpose nuclei dilated: 104,307; Cellpose cell: 87,046; Voronoi: 106,227; Watershed: 105,527; Baysor: 177,437; Baysor (no prior): 191,698), and ranges from the first to third quartile with the median as the horizontal line. The boxplot's lower whisker extends 1.5 times the interquartile range below the first quartile, while the upper whisker extends 1.5 times the interquartile range above the third quartile. **d** Gene-level quality metric represented by a scatter plot of the percentage of cells expressed for each gene in the segmented cells (y-axis) vs. the nuclei (x-axis). **e** Cell morphology metrics represented by the elongation values between the segmented cells (y-axis) and nuclei (x-axis), where each dot represents the average elongation for each cell type and the Pearson correlation between the elongation values of nuclei and segmented cells is noted in the top left corner. **f** Scatter plot between correlation the elongation values of nuclei and segmented cells (y-axis) and average total number of transcripts per cell (x-axis) based on average expression. Source data are provided as a Source Data file.

cell type proportion between the segmented cells generated by BID-Cell and Chromium (cor = 0.95). BIDCell also has a higher presence of positive markers and a lower presence of negative markers in large cells (Fig. 3d and Supplementary Fig. 13), demonstrating an improvement in the expression purity of segmentation.

In category III of CellSPA, we investigate the potential contamination between neighbouring cells by comparing the percentage of B cells that expressed negative markers, such as CD3D and CD3E, which are positive T cell markers but are considered negative markers in B cells. The presence of T cell marker genes in B cells suggests potential contamination during the cell segmentation process. Fig. 3e and Supplementary Fig. 14 indicate that BIDCell showed the smallest percentage of contamination cells, indicating its ability to reduce contamination in a densely populated region.

Lastly, we investigate the spatial diversity by examining the association between the cell type composition and the various cell level characteristics of spatial local regions. Here, we expect the region with a diverse composition of cell types would have a high variety of cell sizes and morphologies. We first divide the image into several local regions and then quantify the diversity of the cell type composition of a region using entropy (Fig. 3f). As shown in Fig. 3g and Supplementary Fig. 15, we find that BIDCell achieves a higher correlation of the coefficient of variation of the cell-level characteristics (the total transcripts, the total genes expressed and cell area) with the cell type entropy compared to the other methods. Similarly, we observe that the variety of cell elongation in BIDCell is highly correlated with the proportion of fibroblasts, one of the dominant cell types in the data (Fig. 3h).

Together, with a comprehensive benchmarking using CellSPA, we demonstrate that the BIDCell segmentation achieves a better balance between high cell expression purity and a large cell body compared to the other state-of-the-art methods, which capture a more diverse range of cell morphologies and provide the potential for a more accurate representation of the topographic context of neighbouring cellular interactions.

## BIDCell is replicable and generalisable to multiple SST platforms

As an additional sensitivity analysis to the ablation study, we evaluated the replicability of BIDCell. We compared the results between the two replicated studies (Xenium-BreastCancer1 and Xenium-BreastCancer2). Figure 3i displays images of the two replicates, with corresponding cell types highlighted in Fig. 3j (left panel). The results are very similar, demonstrating that BIDCell is replicable. The tSNE plot in Fig. 3j (right panel) shows a well-mixed population of cells between the two replicated studies. The high correlation of the cell morphology metrics of segmented cells from BIDCell between the two replicates further confirm the replicability of our method (Supplementary Fig. 16).

We demonstrate the generalisability of BIDCell to other SST platforms and tissue types by applying BIDCell to data generated by CosMx from NanoString (Fig. 4a–c, Supplementary Figs. 17, 18) and MERSCOPE data from Vizgen (Fig. 4d–f, a–c, Supplementary Figs. 19, 20). In particular, we observed that BIDCell had a lower percentage of B cells expressing negative markers (markers indicating contamination) for the CosMx-Lung data (Fig. 4c), suggesting more accurate cell segmentation. Additionally, in MERSCOPE-Melanoma data, regions with more diverse cell types corresponded to more diverse cell type characteristics (Fig. 4f). Furthermore, we also applied BIDCell to Stereo-seq from BGI (Supplementary Fig. 21). We have now demonstrated the applicability of BIDCell on data from four major platforms, and from five different tissue types. We believe that our method has the flexibility and generalisability to other data from other SST platforms and tissues.

## Accurate cell segmentation can reveal region-specific subtypes among neuronal cells

To further assess the performance of BIDCell in accurately segmenting closely packed cells, we performed an evaluation on another case study from Xenium-MouseBrain data. The hippocampus is critical for learning and memory[20], and the tripartite synapses formed between the dentate gyrus and cornu ammonis (CA) have been well studied[21]. Because of the density of pyramidal neurons within the CA region, we asked whether or not BIDCell could accurately distinguish CA1, 2, and 3 from one another. Figure 5a, b show the spatial image and highlight the neuronal cell type and neuronal regions using scClassify trained existing sequencing data (Table 1). Fig. 5c compares the segmentation pattern obtained using 10x vs. BIDCell. Note that BIDCell generates a more finely textured and tighter pattern of cells than 10x, and the output more closely resembles the pattern seen in Fig. 5a. The superior performance of BIDCell is further confirmed by the evaluation metrics. With similar size of the segmented cells with 10x (Supplementary Fig. 22), BIDCell achieves a higher similarity with scRNA-seq and expression purity score (Fig. 5d–e, Supplementary Fig. 23). BIDCell can identify neuronal subtype markers that distinguish granule neurons in the dentate gyrus (Prox1) from pyramidal neurons in CA1-3 (Neurod6) ([22]; Fig. 5f). Furthermore, it is able to spatially subdivide pyramidal neurons in the CA region despite their close proximity to one another. Fig. 5f shows the expression patterns of Wfs1 in CA1[23], Necab2 in CA2[24] and Slit2 in CA3[25], consistent with prior studies. Interestingly, we found a new gene (Cpne8) that is enriched in CA1, consistent with in situ data from the Allen Brain Atlas and illustrates BIDCell's capacity for biological discovery.

## Discussion

Here we presented BIDCell, a method for cell segmentation in subcellular spatially resolved transcriptomics data. BIDCell leverages DL with its biologically-informed loss functions that allow the model to self-learn and capture both cell type and cell shape information, while optimising for cell expression purity. Its default components (such as the backbone architecture and use of cell type profiles) may be exchanged for other architectures and Atlas datasets. We have demonstrated the effectiveness of BIDCell by comparing it to state-of-the-art methods and have shown that BIDCell provides better cell body delineation. Moreover, our flexible approach can be applied to different technology platforms, and different gene panels. Our study highlights the potential of BIDCell for accurate cell segmentation and its potential impact on the field of subcellular spatially resolved transcriptomics.

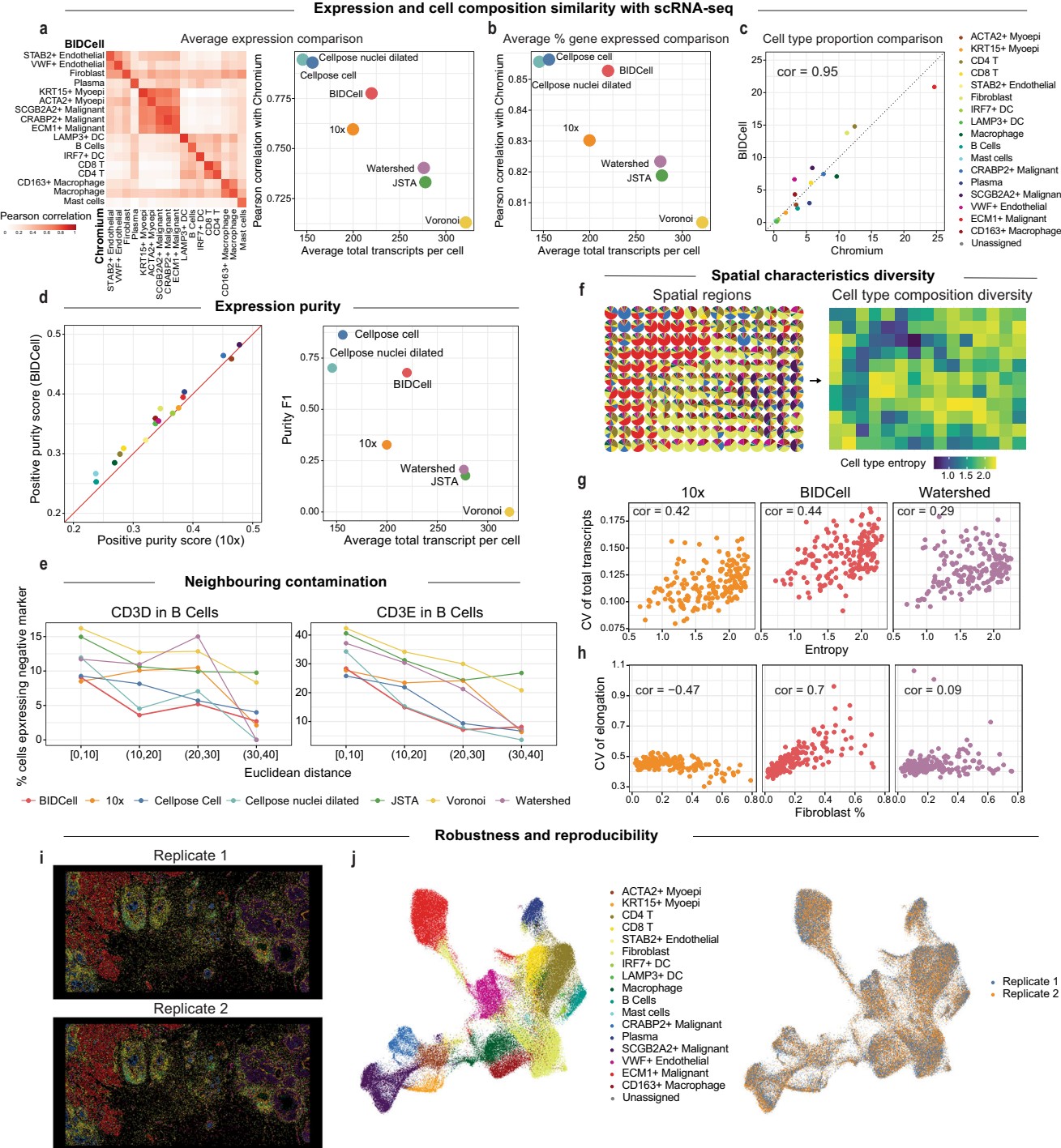

**Fig. 3 | CellSPA graphical representation of comparison study using Xenium-BreastCancer. a** Correlation heatmap of average expression between segmented cells from BIDCell (*y*-axis) and expression from Chromium data (*x*-axis) [left]. Scatter plot between correlation with Chromium expression (*y*-axis) and average total number of transcripts per cell (*x*-axis) based on average expression [right]. Each dot represents a different method. **b** Scatter plot between correlation with Chromium expression (*y*-axis) and average total number of transcripts per cell (*x*-axis), where each dot represents a different method. **c** Scatter plot between BIDCell (*y*-axis) and expression from Chromium data (*x*-axis) based on the cell type proportion extracted from each of the methods. **d** Scatter plot showing the expression between the F1 score for positive markers in BIDCell (*y*-axis) and in 10x segmentation (*x*-axis) [left], and scatter plot showing the purity F1 score against the average total transcripts per cell [right]. Each dot represents a method. **e** Line plots showing the percentage of B cells expressing the unwanted T cell marker CD4, CD8A, and CD8B against its distance from the nearest T cell, where the B cells are grouped by distance ranges. A lower percentage is better, and each line represents a different method. **f**–**h** Spatial characteristics diversity. **f** indicates the local spatial regions being divided in the images where the left panel indicates the cell type proportions of each local region and the right panel indicates the cell type entropy of the local region. **g** Scatter plots showing the association between the cell type entropy and the coefficient of variation of the total transcripts of three methods: 10x, BIDCell, and Watershed, where each dot represents each local region shown in (**f**). **h** Scatter plots showing the association between the coefficient of variation of elongation and proportion of fibroblasts in the data. **i** Spatial imaging of two replicates in Xenium-BreastCancer, where each dot represents the segmented cells coloured by the annotated cell type. **j** UMAP plots of the two replicates, coloured by cell type [left] and replicate [right]. Source data are provided as a Source Data file.

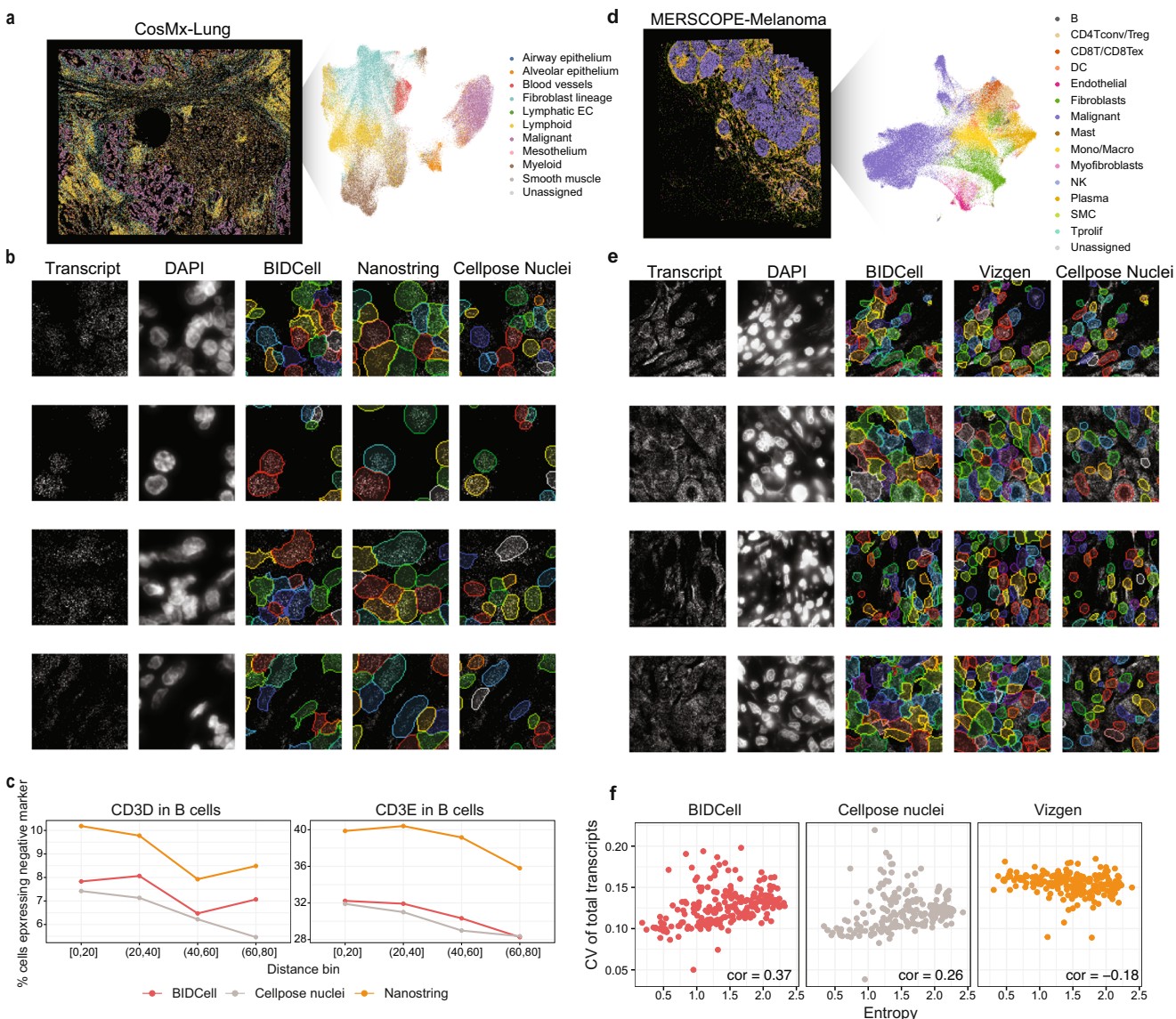

**Fig. 4 | Generalisability of BIDCell. a** CosMx-Lung image with UMAP plot highlighting different cell types. **b** Comparative illustration of the predictions from BIDCell, NanoString and Cellpose nuclei for CosMx-Lung. **c** Line plots showing the percentage of B cells expressing the unwanted T cell marker CD4, CD8A, and CD8B against its distance from the nearest T cell, where the B cells are grouped by the distance ranges. A lower percentage is better, and each line represents a different method with BIDCell (red), NanoString (orange), and Cellpose nuclei (grey).

**d** MERSCOPE-Melanoma image with UMAP highlighting different cell types. **e** Comparative illustration of the predictions from BIDCell, Vizgen and Cellpose nuclei for MERSCOPE-Melanoma. **f** Scatter plot showing the coefficient of variation of the total number of genes against cell type entropy in a given region for cells segmented from BIDCell [left], nuclei cells [middle], and cells segmented from Vizgen [right]. Source data are provided as a Source Data file.

The typical approach to leverage advancements in DL relies on ground truth to guide models to learn relationships between inputs and outputs. However, manual annotation of individual pixels is unattainable for SST that contain hundreds of molecular units per pixel, given the time and effort of manual labour. Further, we have shown (e.g., with Cellpose) that models pretrained on other imaging modalities do not transfer well to SST images. BIDCell innovates through its integrated loss functions that inject biological knowledge of cell morphology and expressions, to allow the model to self-learn from the given spatial transcriptomic and DAPI images, and produce superior visual and quantitative performance compared to previous methods. Our loss functions also allow BIDCell to be broadly applicable across diverse tissue types and various SST platforms. Therefore, BIDCell can facilitate faster research outputs and new discoveries.

Establishing an easy-to-use evaluation system is crucial for promoting reproducible science and transparency, as well as facilitating

further methods development. In CellSPA, we have extended beyond a single accuracy metric and introduced metrics that represent important downstream properties or biological characteristics recognised by scientists. This concept of evaluation by human-recognised criteria is also discussed by the computer vision community as "empirical evaluation"[26]. Another aspect that is often overlooked is related to the practical establishment of benchmarking studies. As benchmarking studies gain recognition, they can be time-consuming due to challenges with software versioning and different operating systems, and different methods may require varying degrees of ease of use and time to adjust the code for comparison. The CellSPA tool is available as a R package with all necessary dependencies, simplifying its installation and usage on local systems, and promoting reproducible science and transparency. Rather than generating a comprehensive comparison of existing methods, which can quickly become outdated, evaluation metrics are generated to

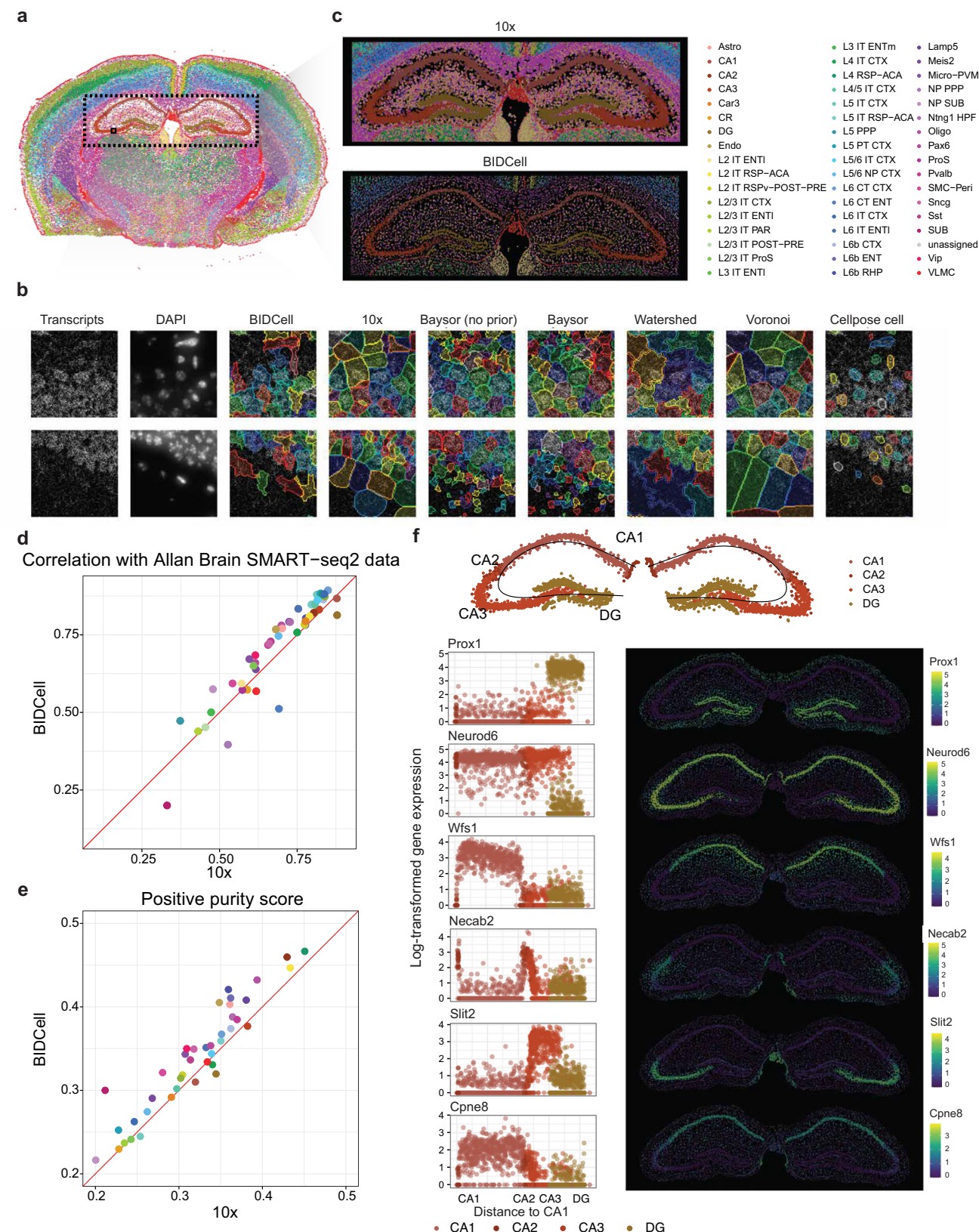

allow new methods to be compared to a database of existing methods, without the need to re-implement a large collection of methods. This approach reduces redundancy, allows for direct comparison with state-of-the-art methods, and saves time and effort. Examples of this approach include those for cell deconvolution[27] and simulation methods[28].

A comprehensive evaluation framework is vital when comparing diverse segmentation approaches in the absence of a ground truth. It is important to recognise that different segmentation approaches may purposefully have different priorities and outcomes. For example, a segmentation approach such as a seeded Voronoi tessellation will identify larger cells than a fixed expansion around the nuclei, such as

**Fig. 5 | Assessment using Xenium-MouseBrain data. a** Spatial image highlighting the cell type and neuronal regions using scClassify trained on SMART-seq2 data. **b** Comparative illustration of the predictions from BIDCell and other methods. **c** Hippocampus cell segmentation region by 10x [top] and BIDCell [bottom]. **d** Scatter plot showing the Pearson correlation with SMART-seq2 data between 10x and BIDCell for each cell type, where each dot is coloured by the cell type with the same colours as the legend in (**c**). **e** Scatter plot showing the positive purity score between 10x and BIDCell for each cell type, where each dot is coloured by the cell type. **f** The top panel indicates the neurons in the hippocampus region (CA1-CA3, DG) and the bottom panels are 6 x 2 panels showing the five distinct spatial regions with different neuronal markers in the hippocampal regions. From top to bottom, *Prox1* was expressed only in DG, *Neurod6* was expressed in all CA regions, *Slit2* was expressed in CA3, *Necab2* was expressed in CA2, and *Wfs1* and *Cpne8* were expressed in CA1. Source data are provided as a Source Data file.

Cellpose cell. The former will typically assign more transcripts and produce a denser map of which cells are touching. In contrast, the latter may produce more homogenous profiles of the cells with fewer assigned molecules and tighter cell boundaries, limiting its capability to estimate physical cell interactions. While achieving more homogenous cell bodies is desired, it can also result from the arbitrary over-segmentation of nuclei. This emphasises that the use of employing a variety of metrics to quantify segmentation performance enables a systematic assessment and reveals the desirable properties of each approach.

Cells have a three-dimensional structure, thus analyses in a two-dimensional perspective may achieve limited representation. BIDCell can be further adapted (e.g., via its cell-calling loss) to incorporate cell membrane markers to enhance segmentation. In MERSCOPE data that display cell membrane markers, there is a percentage (25%) of cells that lack nuclei in their segmentation, likely due to being elongated melanocytes or fibroblasts in a section without a nucleus. While platforms like MERSCOPE can utilise cell membrane markers as cell masks to perform cell segmentation, it is necessary to conduct further research to understand whether a cell's slicing affects the measurement of expression in tissues. Similarly, in the nervous system, a future challenge will be to accurately identify and segment dendritic and axon morphologies. Like melanocytes and fibroblasts, the varied and elongated nature of these cell morphologies will make it challenging to accurately identify cell boundaries in the absence of nearby nuclei. Because of these difficulties, most approaches may instead generate similar results between the segmentation of the whole cell and the corresponding segmentation of the cell nuclei.

In conclusion, the development of subcellular spatial transcriptomics technologies is revolutionising molecular biology. We have introduced a deep learning approach that does not require ground truth supervision and incorporates prior biological knowledge by leveraging the myriad of single-cell datasets in Atlas databases. We illustrate that our BIDCell method outperforms the current state-of-the-art cell segmentation methods, and we are able to uncover region-specific subtypes in the brain with explicit highlighting of cell bodies and boundaries. Furthermore, recognising the importance of evaluation, we developed CellSPA, a Cell Segmentation Performance Assessment framework, that covers a wide variety of metrics across five complementary categories of cell segmentation characteristics.

## Methods
### Datasets and preprocessing
We used publicly available data resources from three different SST commercial platforms (10 × Genomics Xenium, NanoString CosMx, and Vizgen MERSCOPE), and sequencing data from Human Cell Atlas.

**Subcellular spatial transcriptomics data.** For all datasets and for each gene, detected transcripts were converted into a 2D image where the value of each pixel represents the number of detected transcripts at its location. The images were combined channel-wise, resulting in an image volume $X \in \mathbb{R}^{H \times W \times n_{genes}}$, where $H$ is the height of the sample, $W$ is the width of the sample, and $n_{genes}$ is the number of genes in the panel.

**(i) Xenium-BreastCancer1 and Xenium-BreastCancer2.** The Breast Cancer datasets included in this study were downloaded from https://www.10xgenomics.com/products/xenium-in-situ/preview-dataset-human-breast (accessed 9 Feb 2023), and included two replicates. Low-quality transcripts for 10 × Genomics Xenium data with a phred-scaled quality value score below 20 were removed, as suggested by the vendor[1]. Negative control transcripts, blanks, and antisense transcripts were also filtered out. This resulted in 313 unique genes with the overall pixel dimension of the images being $5475 \times 7524 \times 313$ for Xenium breast cancer replicate 1 (Xenium-BreastCancer1) and $5474 \times 7524 \times 313$ for Xenium breast cancer replicate 2 (Xenium-BreastCancer2).

**(ii) Xenium-MouseBrain.** The Mouse Brain data included in this study was downloaded from https://www.10xgenomics.com/resources/datasets/fresh-frozen-mouse-brain-replicates-1-standard (accessed 14 Feb 2023) and were processed following the steps in (i). There were 248 unique genes, and the resulting size of the image was $7038 \times 10,277 \times 248$ pixels.

**(iii) CosMx-Lung.** The CosMx NSCLC Lung dataset included in this study was downloaded from https://nanostring.com/products/cosmx-spatial-molecular-imager/nsclc-ffpe-dataset/ (accessed 24 Mar 2023). We used data for Lung5-1, which comprised 30 fields of view. Transcripts containing "NegPrb" were removed, resulting in 960 unique genes and an overall image dimension of $7878 \times 9850 \times 960$ pixels.

**(iv) MERSCOPE-Melanoma.** The MERSCOPE melanoma data included in this study were downloaded from https://info.vizgen.com/merscope-ffpe-solution (for patient 2, accessed 26 Mar 2023). Transcripts with "Blank-" were filtered out, resulting in 500 unique genes and an image with $6841 \times 7849 \times 500$ pixels.

**(v) Stereo-seq-MouseEmbryo.** The Stereo-seq data used in this study, including the DAPI image and detected gene expressions (bin 1), were downloaded from https://db.cngb.org/stomics/mosta/download/ for sample E12.5_E1S3. Stereo-seq data contains a far greater number of genes compared to Xenium, CosMx, and MERSCOPE. For efficiency, we selected a panel of 275 highly variable genes (HVGs) as the input to BIDCell. The HVGs are the common genes of the top 1000 HVGs from both Stereo-seq data and the single-cell reference data.

**Nuclei segmentation.** DAPI images were directly downloaded from the websites of their respective datasets. In cases where the maximum intensity projection (MIP) DAPI image was not provided, we computed the MIP DAPI by finding the maximum intensity value for each *(x,y)* location for each stack of DAPI. DAPI images were resized to align with the lateral resolutions of spatial transcriptomic maps using bilinear interpolation. Nuclei segmentation was performed on the MIP DAPI using the pretrained Cellpose model with automatic estimation of nuclei diameter[5]. We used the "cyto" model as we found the "nuclei" model to undersegment or omit a considerable number (e.g., 21k for Xenium-BreastCancer1) of nuclei given the same MIP DAPI image, which is consistent with another study[29]. Other nuclei segmentation methods may be used with BIDCell as our framework is not limited to Cellpose.

**Transcriptomics sequencing data.** We used five publicly available single-cell RNA-seq data collections as references to guide the cell segmentation in BIDCell and evaluation with CellSPA. For the reference

data with multiple datasets, we constructed cell-type specific profiles by aggregating the gene expression by cell type per dataset.

**(i) TISCH-BRCA.** The reference for Xenium-BreastCancer used in BID-Cell was based on 10 single-cell breast cancer datasets downloaded from The Tumor Immune Single Cell Hub 2 (TISCH2)[17] from http://tisch.comp-genomics.org/gallery/?cancer=BRCA&species=Human, which contains the gene by cell expressions and cell annotations of the data. We used the "celltype major lineage" as the cell type labels. We combined the "CD4Tconv" and "Treg" as "CD4Tconv/Treg" and "CD8T" and "CD8Tex" as "CD8T/CD8Tex", which results in 17 cell types in total.

**(ii) Chromium-BreastCancer.** To evaluate the performance of Xenium-BreastCancer, we downloaded the Chromium scFFPE-seq data from the same experiment from https://www.10xgenomics.com/products/xenium-in-situ/preview-dataset-human-breast (accessed 22 March 2023), which contains 30,365 cells and 18,082 expressed genes. We then performed Louvain clustering on the k-nearest neighbour graph with $k = 20$, based on the top 50 principal components (PCs) to obtain 22 clusters. We then annotated each cluster based on the markers and annotation provided in the original publication[1].

**(iii) Allen Brain Map.** The reference for Xenium-MouseBrain data was based on Mouse Whole Cortex and Hippocampus SMART-seq data downloaded from https://portal.brain-map.org/atlases-and-data/rnaseq/mouse-whole-cortex-and-hippocampus-smart-seq, which contains both gene by cell expressions and cell annotations of the data. We used the cluster annotation from "cell_type_alias_label" as the cell type labels and combined some of the labels with a small number of cells. For example, we combined all "Sst" subtypes as "Sst" and all "Vip" subtypes as "Vip", which results in 59 cell types in total.

**(iv) HLCA and TISCH-NSCLC.** The reference for CosMx-Lung for both BIDCell and CellSPA was based on Human Lung Cell Atlas (HLCA)[30], provided in the "HLCA_v1.h5ad" file from https://beta.fastgenomics.org/p/hlca, including both gene expressions and cell type annotations of the data. We used "ann_finest_level" as cell type labels, which contained 50 cell types in total.

As HLCA only contains single-cell datasets from non-cancer lung tissue, we complemented the reference data with malignant cells provided in TISCH2, where we downloaded 6 single-cell NSCLC datasets with tumour samples from http://tisch.comp-genomics.org/gallery/?cancer=NSCLC&species=Human. We only included the cells labelled as malignant cells in the reference.

**(v) TISCH-SKCM.** The reference for MERSCOPE-Melanoma for both BIDCell and CellSPA was based on 10 single-cell melanoma datasets downloaded from TISCH2 from http://tisch.comp-genomics.org/gallery/?cancer=SKCM&species=Human, which contains the gene by cell expressions and cell annotations of the data. We used the "celltype major lineage" as the cell type labels. We combined the "CD4Tconv" and "Treg" as "CD4Tconv/Treg" and "CD8T" and "CD8Tex" as "CD8T/CD8Tex", which resulted in 15 cell types in total.

**(vi) Mouse Embryo reference.** The reference for Stereo-seq-MouseEmbryo was downloaded from GEO database under accession code: GSE119945[31], which contain both counts and cell type annotation data. The E12.5 data was then used as reference.

### Biologically-informed deep learning-based cell segmentation (BIDCell) overview

BIDCell is a self-supervised deep learning framework that computes biologically-informed loss functions to optimise learnable parameters for the prediction of cell segmentation masks for spatial transcriptomic data. BIDCell uses three types of data: (i) spatial transcriptomic maps of genes, (ii) corresponding DAPI image, and (iii) average gene expression profiles of cell types from a reference dataset, such as the Human Cell Atlas. A major innovation in developing BIDCell is the use of biologically-informed prior knowledge via the SSL paradigm to enable DL models to learn complex structures in SST data, to derive cell segmentations that are visually more realistic and capture better expression profiles.

The BIDCell framework has the following four key characteristics:

- BIDCell predicts diverse cell shapes for datasets containing various cell types to better capture cell expressions (see section Elongated and non-elongated shapes).
- BIDCell uses positive and negative markers from sequencing data to enhance the guidance for learning relationships between spatial gene expressions and cell morphology in the form of cell segmentations (see section Positive and negative cell-type markers).
- BIDCell is parameterised by a deep learning architecture that learns to segment cells from spatial transcriptomic images (see section Deep learning-based segmentation).
- BIDCell uses biologically-informed, self-supervised loss functions to train the deep learning architecture without the need for manual annotations and better capture cell expressions (see section BIDCell training and loss functions).

**Elongated and non-elongated shapes.** BIDCell is capable of generating cell segmentations that exhibit different morphologies for different cell types, rather than assume a generally circular profile for all cell types. In particular, BIDCell can distinguish between cell types that typically appear more elongated, such as fibroblasts and smooth muscle cells, and those that are typically more rounded or circular, such as B cells. Elongated cell types can be directly specified for each tissue sample as desired, based on existing biological knowledge.

We used the expression within the nuclei (see section Nuclei segmentation) of cells to perform an initial classification of elongated and non-elongated cell types. Transcripts were mapped to nuclei using nuclei segmentations, and the Spearman correlation was computed between nuclei expression profiles and reference cell types of the Human Cell Atlas. Nuclei were classified as the cell type with which it was most highly correlated to. This initial classification coupled with the eccentricity of the nuclei were used to inform the cell-calling loss function (described in section Cell-calling loss) to produce segmentation morphologies with more variation that are more appropriate for different cell types. We considered epithelial cells, fibroblasts, myofibroblasts, and smooth muscle cells to be elongated for samples of breast cancer and melanoma. Endothelial cells, fibroblasts, myofibroblasts, fibromyocytes, and pericytes were deemed elongated for NSCLC. We considered all cell types in the mouse brain sample to be elongated.

**Positive and negative cell-type markers.** BIDCell learns relationships between the spatial distribution of gene expressions and cell morphology in the form of cell segmentations. This relationship can be enhanced by incorporating biological knowledge in the form of cell-type markers, specially, the genes that are typically more expressed (positive markers) and less expressed (negative markers) in different cell types, which allows BIDCell to predict segmentations that lead to more accurate cell expression profiles. Cell-type marker knowledge is drawn from the Human Cell Atlas, which allows BIDCell to be applied without requiring a matched single-cell reference for the same sample of interest. Markers were incorporated into BIDCell through our positive and negative marker losses (described in section Positive and negative marker losses).

**Deep learning-based segmentation.** BIDCell is parameterised by a set of learnable parameters $\theta$ of a deep learning segmentation model. We

used the popular UNet 3+[19] as the backbone of our framework to perform cell segmentation by predicting the probability of cell instances at each pixel. This architecture may be swapped out for other segmentation architectures. UNet 3+ was originally proposed for organ segmentation in computed tomography (CT) images. It was built on the original U-Net[12] and incorporated full-scale skip connections that combined low-level details with high-level features across different scales (resolutions). UNet 3+ comprised an encoding branch and decoding branch with five levels of feature scales. We did not adopt the deep supervision component proposed by UNet 3+, and instead only computed training losses at the lateral resolution of the original input.

### Input

The input to the UNet 3+ model was a cropped multichannel spatial transcriptomic image $\mathbf{x} \in \mathbb{R}^{h \times w \times n_{genes}}$, where $n_{genes}$ represents the channel axis corresponding to the total number of genes in the dataset, $h$ is the height of the input patch, and $w$ is the width of the input patch. Prior to being fed into the first convolutional layer, the input was reshaped to $[n_{cells}, n_{genes}, h, w]$, effectively placing $n_{cells}$ in the *batch size* dimension. In this way, all the cells in a patch were processed simultaneously, and the model could flexibly support an arbitrary number of cells without requiring extra padding or preprocessing. $n_{cells}$ was determined by the corresponding patch of nuclei to ensure consistency with predicted cell instances. Input volumes that were empty of nuclei were disregarded during training and yielded no cells during prediction.

### Output and segmentation prediction

The softmax function was applied to the output of UNet 3+ to yield probabilities of foreground and background pixels for each cell instance. This produced multiple probabilities for background pixels (i.e., $n_{cells}$ probabilities per pixel for a patch containing $n_{cells}$), due to the placement of cell instances in the *batch size* dimension. These probabilities were aggregated by averaging across all the background predictions per pixel. The *argmax* function was applied pixel-wise to the foreground probabilities for all cells and averaged background probabilities. This produced a segmentation map corresponding to the object (cell instance or background) with the highest probability at each pixel.

### Morphological processing

The initial segmentation output by the deep learning model was further refined to ensure pixel connectivity within each cell (i.e., all the sections of the cell were connected). The process involved standard morphological image processing techniques to each cell, including dilation, erosion, hole-filling, and removal of isolated islands, while ensuring that the nucleus was captured. First, dilation followed by erosion were applied using a $5 \times 5$ circular kernel with two iterations each. Hole-filling was then carried out on the cell section with the largest overlap with the nucleus. Any remaining pixels initially predicted for the cell that were still not connected to the main cell section were discarded. After morphological processing, the number of transcripts captured within each cell is slightly higher, while purity metrics and correlation with Chromium are the same or slightly higher (Supplementary Fig. 24).

### Mapping transcripts to predicted cells

The detected transcripts were mapped to cells using the final predicted segmentations. The segmentation map was resized back to the original pixel resolution using nearest neighbour interpolation. Transcripts located in the mask of a cell were added to the expression profile of the cell. This produced a gene-cell matrix $n_{cells} \times n_{genes}$, which was used for performance evaluation and downstream analysis.

**BIDCell training and loss functions.** The BIDCell framework combines several loss functions that automatically derive supervisory signals from the input data and/or predicted segmentations at each step of the training process. This approach to learning is a core aspect of SSL[32]. Furthermore, the modular and additive design of the loss

functions allows each loss to be swapped out with alternative approaches to compute training signals. The SSL label describes the ability of the framework to automatically learn relationships between gene expressions and cell morphology from its inputs.

Our approach for learning the parameters $\theta$ of the segmentation model relies on minimising a total of 6 loss functions that we propose with our framework. Some of the losses effectively increase the number of pixels predicted for a cell, while others reduce the size of its segmentation. The nuclei encapsulation, cell-calling, over-segmentation, and overlap losses guide the basic morphology of cells. The positive and negative marker losses refine the cell morphologies learned through the other loss functions, by further guiding the model to learn biologically-informed relationships between gene expressions and cell morphology. This is reminiscent of the pretext and downstream (fine-tuning) stages commonly encountered in SSL, where the pretext task aids the model to learn better representations or intermediate weights, while the fine-tuning task refines the weights and further improves performance for a particular prediction task. Taken together, the losses ensure that the segmentation model learns relationships between spatially-localised, high-dimensional gene expression information and the morphology of individual cells.

### (A) Nuclei encapsulation loss

The segmentation of a cell must contain all the pixels of the cell's nucleus. Additionally, the expressed genes in nuclei can guide the model to learn which genes should be predicted within cells. Hence, we included a loss function $L_{ne}$ that incentivises the model to learn to correctly predict nuclei pixels:

$$L_{ne}(\mathbf{x_{nuc}}, \hat{\mathbf{y}}) = -\mathbf{x_{nuc}} \log(\hat{\mathbf{y}}) - (1 - \mathbf{x_{nuc}}) \log(1 - \hat{\mathbf{y}}), \qquad (1)$$

where $\mathbf{x_{nuc}}$ is the binary nucleus segmentation mask, and $\hat{\mathbf{y}}$ is the predicted segmentation for all cells of the corresponding training patch.

### (B) Cell-calling loss

The aim of the cell-calling loss was to increase the number of transcripts assigned to cells. We also designed the cell-calling loss to allow BIDCell to capture cell-type specific morphologies. Unique expansion masks $\mathbf{e_c} \in \{0, 1\}^{h \times w}$ were computed for each cell based on the shape of its nucleus and whether its nucleus expression profile was indicative of an elongated cell type. The expansion mask of a non-elongated cell was computed by applying a single iteration of the morphological dilation operator with a circular kernel of $20 \times 20$ pixels to its binary nucleus mask.

The expansion mask of an elongated cell was computed based on the elongation of its nucleus, defined as the eccentricity of an ellipse fitted to its nucleus mask:

$$ecc = \sqrt{1 - \frac{b^2}{a^2}}, \qquad (2)$$

where $a$ represents the length of the major axis, and $b$ is the length of the minor axis.

We found that elongated cell types tended to have nuclei with higher eccentricity (Supplementary Fig. 1). Hence, the eccentricity of a nucleus could serve as a proxy for the shape of its cell via an elongated expansion mask. We computed each cell-specific elongated expansion mask using an elliptical dilation kernel applied to the nucleus. The horizontal and vertical lengths of the elliptical kernel were computed by:

$$l_h = \alpha \times ecc_{nuc} \times l_t, \qquad (3)$$

$$l_v = \begin{cases} l_t - l_h, & \text{if } l_t - l_h > l_{vm} \\ l_{vm}, & \text{otherwise} \end{cases} \qquad (4)$$

where $\alpha$ is a scaling factor set to 0.9, $ecc_{nuc}$ is the eccentricity of the nucleus, $l_t$ is the sum of $l_h$ and $l_v$, which was set to 60 pixels, and $l_{vm}$ is the minimum vertical length, which was set to 3 pixels. These values were selected based on visual inspection (e.g., the cells appear reasonably sized), and were kept consistent across the different elongated cell types and datasets used in this study. The elliptical dilation kernel was rotated to align with the nucleus and applied to the nucleus mask to produce the elongated expansion mask of the cell.

The expansion masks were used in our cell-calling loss function that was minimised during training:

$$L_{cc}(\mathbf{e},\hat{\mathbf{y}}) = \frac{1}{M}\sum_c^M -\mathbf{e_c}\log(\hat{\mathbf{y}}_c) - (1-\mathbf{e_c})\log(\mathbf{1}-\hat{\mathbf{y}}_c), \quad (5)$$

where $\mathbf{e_c}$ is the expansion mask and $\hat{\mathbf{y}}_c$ is the predicted segmentation of cell $c$ of $M$ cells in an input patch.

### (C) Over-segmentation loss

We introduced the over-segmentation loss to counter the cell size-increasing effects of the cell-calling loss to prevent the segmentations becoming too large and splitting into separate segments. This loss function elicited a penalty whenever the sum of cytoplasmic predictions exceeded the sum of nuclei predictions for a cell in a given patch:

$$p_{nuc,c} = \sum_i \sum_j \sigma(\hat{q}_{ijc} x_{nuc,ij} - 0.5), \quad (6)$$

$$p_{cyto,c} = \sum_i \sum_j \sigma(\hat{q}_{ijc}(1 - x_{nuc,ij}) - 0.5), \quad (7)$$

$$L_{os} = \begin{cases} \frac{1}{M}\sum_c^M (p_{cyto,c} - p_{nuc,c}), & \text{if } \sum_c^M (p_{cyto,c} - p_{nuc,c}) > 0 \\ 0, & \text{otherwise} \end{cases} \quad (8)$$

where for cell $c$ at pixel $(i,j)$, $\hat{q}_{ijc}$ is the predicted foreground probability for cell $c$, $x_{nuc,ij} \in \{0,1\}$ is the binary nucleus mask, and $\sigma$ is the sigmoid function. $L_{os}$ was normalised by number of cells $M$ to aid smooth training.

### (D) Overlap loss

Cells are often densely-packed together in samples of various human tissues. This poses a challenge to segmentation models in predicting clear boundaries and coherent segmentations for neighbouring cells without overlap. We introduced the overlap loss to penalise the prediction of multiple cells occurring at each pixel:

$$s_{ov,ij} = -(1 - x_{nuc,ij}) + \sum_c^M \sigma(\hat{q}_{ijc}(1 - x_{nuc,ij}) - 0.5), \quad (9)$$

$$L_{ov} = \begin{cases} \frac{\sum_i\sum_j(s_{ov,ij})}{Mhw}, & \text{if } s_{ov} > 0 \\ 0, & \text{otherwise} \end{cases} \quad (10)$$

$L_{ov}$ was normalised by number of cells $M$, and the lateral dimensions $h$ and $w$ of the input to aid smooth training.

### (E) Positive and negative marker losses

The purposes of our positive and negative marker losses were to encourage the model to capture pixels that contained positive cell-type markers, and penalise the model when segmentations captured pixels that contained negative cell-type markers for each cell. The marker losses refine the initial morphology learned through the other loss functions, by further guiding the model to learn biologically-informed relationships between gene expressions and cell morphology.

The positive and negative markers for the training loss were those with expressions in the highest and lowest 10 percentile for each cell type of a tissue sample. In our experiments, we found that a higher number of positive markers tended to increase the size of predicted cells as the model learns to capture more markers, and vice versa. We found that removing positive markers that were common to at least a third of cell types in each tissue type was appropriate across the different datasets for training.

The one-hot encoded lists of positive and negative markers of the cell type for cell $c$ were converted into sparse maps $\mathbf{m_{pos,c}} \in \{0,1\}^{h\times w}$ and $\mathbf{m_{neg,c}} \in \{0,1\}^{h\times w}$. At each pixel, 0 indicated the absence of all markers, while 1 indicated the presence of any positive or negative marker for its respective map. $\mathbf{m_{pos,c}}$ and $\mathbf{m_{neg,c}}$ were then multiplied element-wise by the expansion mask $\mathbf{e_c}$ to remove markers far away from the current cell. Each marker map was dilated by a $3\times 3$ kernel, which was based on the assumption that pixels in a $3\times 3$ region around each marker were most likely from the same cell. We found this dilation to improve training guidance and segmentation quality, as the maps tended to be quite sparse.

The marker maps were then used to compute the positive and negative marker losses:

$$L_{pos}(\mathbf{m_{pos}},\hat{\mathbf{y}}) = \frac{1}{M}\sum_c^M -\mathbf{m_{pos,c}}\log(\hat{\mathbf{y}}_c) - (1-\mathbf{m_{pos,c}})\log(\mathbf{1}-\hat{\mathbf{y}}_c), \quad (11)$$

$$L_{neg}(\mathbf{m_{neg}},\hat{\mathbf{q}}) = \frac{1}{M}\sum_c^M \sigma(\hat{\mathbf{q}}_c \mathbf{m_{neg,c}} - 0.5) \quad (12)$$

**Total loss**. The model was trained by minimising the sum of all the loss functions over $N$ training patches:

$$\min_\theta \sum_n^N [\lambda_{ne}L_{ne} + \lambda_{cc}L_{cc} + \lambda_{os}L_{os} + \lambda_{ov}L_{ov} + \lambda_{pos}L_{pos} + \lambda_{neg}L_{neg}], \quad (13)$$

where each $\lambda$ represents a hyperparameter that scaled its respective $L$. The value of $\lambda$ for all loss functions was set to 1.0 (except for the ablation and lambdas studies); this ensured our losses were not fine-tuned to any particular datasets.

## Practical implementation

**Details**. To address computational efficiency concerns related to memory usage, we partitioned the spatial transcriptomic maps into patches of $48 \times 48 \times n_{genes}$ for input into UNet 3+. BIDCell has been verified for datasets containing up to 960 genes on a 12 GB GPU. It is also important to note that the number of genes primarily affects the weights of the first convolutional layer, thus having a minor impact on memory usage.

The patch-based predictions could result in effects along the patch boundaries such as sharp or cut-off cells. When dividing the transcriptomic maps into patches, we create two sets of patches of the same lateral dimensions with an overlap equal to half the lateral size of the patches. The predictions for the patches were combined (see Supplementary Fig. 25), without additional operations to resolve potential disagreement between predictions of the two sets. Only patches from the first set (no overlaps) were selected during training, while all patches were used during inference.

One image patch was input into the model at one time, though batch size was effectively $n_{cells}$ due to reshaping (see section Deep learning-based segmentation-Input). Neither normalisation nor standardisation were applied to the input image patches, such that the pixels depicted raw detections of transcripts.

The model was trained end-to-end from scratch for 4000 iterations (i.e., using 4000 training patches). This amounted to a maximum of 22% of the entire image, thereby leaving the rest of the image unseen

by the model during inference. Weights of the convolutional layers were initialised using He et al.'s method[33]. We employed standard on-the-fly image data augmentation by randomly applying a flip (horizontal or vertical), rotation (of 90, 180, or 270 degrees) in the *(x,y)* plane. The order of training samples was randomised prior to training. We employed the Adam optimiser[34] to minimise the sum of all losses at a fixed learning rate of 0.00001, with a first moment estimate of 0.9, second moment estimate of 0.999, and weight decay of 0.0001.

**Time and system considerations.** We ran BIDCell on a Linux system with a 12GB NVIDIA GTX Titan V GPU, Intel(R) Core(TM) i9-9900K CPU @ 3.60GHz with 16 threads, and 64GB RAM. BIDCell was implemented in Python using PyTorch. For Xenium-BreastCancer1, which contained 109k detected nuclei, 41M pixels *(x,y)*, and 313 genes, training was completed after approximately 10 minutes for 4000 steps. Inference time was about 50 minutes for the complete image. Morphological processing required approximately 30 min to generate the final segmentation. A comparison of the runtimes between different methods is included in Supplementary Fig. 26.

**Ablation study.** We performed an ablation study to determine the contributions from each loss function and effects of different hyperparameter values (Supplementary Figs. 4, 5). We used Xenium-BreastCancer1 for these experiments. We evaluated BIDCell without each of the different loss functions by individually setting their corresponding weights $\lambda$ to zero. Furthermore, we evaluated different parameterisations of the cell-calling loss. We experimented with different diameters for the dilation kernel for non-elongated cells, including 10, 20, and 30 pixels, and different total lengths of the minor and major axes $l_t$ of the dilation kernel for elongated cells, including 50, 60, and 70 pixels. We also ran BIDCell without shape-specific expansions, thereby assuming a non-elongated shape for all cells.

## Performance evaluation

We compared our BIDCell framework to vendor-provided cell segmentations, and methods designed to identify cell bodies via cell segmentation. Table 2 provides a summary of all methods compared from adapting classical approaches including Voronoi expansion, nuclei dilation, and the watershed algorithm, to recently proposed approaches for SST images including Baysor, JSTA, and Cellpose. Methods that were excluded from the evaluations include those that focus on the assignment of transcripts to cells and do not consider the cell boundaries, underperformance on the public datasets, lack of code and instructions to prepare data into the required formats, and failure of the method to detect any cells (Supplementary Table 1).

**Settings used for other methods.** We used publicly available code for Baysor, JSTA, and Cellpose with default parameters unless stated otherwise. All comparison methods that required nuclei information used identical nuclei as BIDCell, which were detected using Cellpose (v2.1.1) (see Nuclei segmentation).

- Baysor - Version 0.5.2 was applied either without a prior, or with a prior nuclei segmentation with default prior segmentation confidence of 0.2. For both instances, we followed recommended settings[35], including 15 for the minimum number of transcripts expected per cell, and not setting a scale value, since the sample contained cells of varying sizes. We found the scale parameter to have a considerable effect on segmentation predictions, and often resulted in cells with unrealistically uniform appearances if explicitly set.
- JSTA - default parameters were used. We encountered high CPU loading and issues with two regions of Xenium-BreastCancer1, which yielded empty predictions for those regions despite multiple attempts and efforts to reduce input size.

- Cellpose - Version 2.1.1 was applied to the channel-wise concatenated image comprising DAPI as the "nuclei" channel, and sum of spatial transcriptomic maps across all genes as the "cells" channel, using the pre-trained "cyto" model with automatic estimation of cell diameter.
- Voronoi - Classical Voronoi expansion was seeded on nuclei centroids and applied using the SciPy library (v1.9.3).
- Watershed - The watershed algorithm was performed on the sum of transcriptomic maps across all genes. Seeded watershed used nuclei centroids and was applied using OpenCV (v4.6.0).
- Cellpose nuclei dilation - we applied dilation to nuclei masks as a comparison segmentation method. Each nucleus was enlarged by about 1 micron in radius by applying morphological dilation using a 3 × 3 circular kernel for one iteration. Overlaps between adjacent cell expansions were permitted.

**Evaluation metrics and settings.** We introduce the CellSPA framework, that captures evaluation metrics across five complementary categories. A summary of this information is provided in Supplementary Table 2.

### [A] Baseline metrics
**Overall characteristics**
- Number of cells
- Proportion of transcripts assigned

**Cell-level QC metrics**
- Proportion of cells expressing each gene
- Number of transcripts per cell
- Number of genes expressed per cell
- Cell area

$$\text{Density} = \frac{\sum_{i \in I} n_i}{A}, \tag{14}$$

where $\sum_{i \in I} n_i$ represents the sum of all total transcripts over a set $I$, and $A$ represents the cell area.

**Cell morphology metrics**

We evaluated multiple morphology-based metrics and provide diagrammatic illustrations in Supplementary Fig. 27.

- Elongation =

$$\frac{W_{\text{bb}}}{H_{\text{bb}}}, \tag{15}$$

where $W_{\text{bb}}$ represents the width of the bounding box, and $H_{\text{bb}}$ represents the height of the bounding box.

Elongation measures the ratio of height versus the width of the bounding box (Supplementary Fig. 27f). Elongation is insensitive to concave irregularities and holes present in the shape of the cell. The value of this metric will be 1 for a perfect square bounding box. As the cell becomes more elongated the value will either increase far above 1 or decrease far below 1, depending on whether the elongation occurs along the height or width of the bounding box.

- Circularity =

$$\frac{4\pi \times A}{P_{\text{convex}}^2}, \tag{16}$$

where $A$ represents the area, and $P_{\text{convex}}$ represents the convex perimeter.

Circularity measures the area to perimeter ratio while excluding local irregularities of the cell. We used the convex perimeter of the object as opposed to its true perimeter to avoid concave irregularities. The value will be 1 for a circle and decreases as a cell becomes less circular.

• Sphericity =

$$\frac{R_{\mathrm{I}}}{R_{\mathrm{C}}}, \qquad (17)$$

where $R_{\mathrm{I}}$ represents the radius of the inscribing circle, and $R_{\mathrm{C}}$ represents the radius of the circumscribing circle.

Sphericity measures the rate at which an object approaches the shape of a sphere while accounting for the largest local irregularity of the cell by comparing the ratio of the radius largest circle that fits inside the cell (inscribing circle) to the radius of the smallest circle that contains the whole cell (circumscribing circle). The value is 1 for a sphere and decreases as the cell becomes less spherical.

• Compactness =

$$\frac{4\pi \times A}{P_{\mathrm{cell}}^2}, \qquad (18)$$

where $A$ represents the area, and $P_{\mathrm{cell}}$ represents the cell perimeter.

Compactness measures the ratio of the area of an object to the area of a circle with the same perimeter. Compactness uses the perimeter of the cell thus it considers local irregularities in the cell perimeter. A circle will have a value of 1, and the less smooth or more irregular the perimeter of a cell, the smaller the value will be. For most cells the numerical values for compactness and circularity are expected to be similar. Identifying which cells have large differences between these metrics can identify cells with highly irregular perimeters which may be of interest for downstream analysis and quality control for segmentation.

• Convexity =

$$\frac{P_{\mathrm{convex}}}{P_{\mathrm{cell}}}, \qquad (19)$$

where $P_{\mathrm{convex}}$ represents the convex perimeter and $P_{\mathrm{cell}}$ represents the cell perimeter.

Convexity measures the ratio of the convex perimeter of a cell to its perimeter. The value will be 1 for a circle and decrease the more irregular the perimeter of a cell becomes, similar to compactness.

• Eccentricity =

$$\frac{L_{\mathrm{minor}}}{L_{\mathrm{major}}}, \qquad (20)$$

where $L_{\mathrm{minor}}$ represents the length of the minor axis and $L_{\mathrm{major}}$ represents the length of the major axis.

Eccentricity (or ellipticity) measures the ratio of the major axis to the minor axis of a cell. The major axis is the longest possible line that can be drawn between the inner boundary of a cell without intersecting its boundary. The minor axis is the longest possible line can be drawn within the inner boundary of a cell while while also being perpendicular to the major axis. This gives a value of 1 for a circle and decreases the more flat the cell becomes.

• Solidity =

$$\frac{A}{A_{\mathrm{convex}}}, \qquad (21)$$

where $A$ represents the area, and $A_{\mathrm{convex}}$ represents the convex area.

Solidity measures the ratio of the area of a cell to the convex area of a cell. This measures the density of a cell by detecting holes and irregular boundaries in the cell shape. The maximum value will be 1 for a cell with a perfectly convex and smooth boundary and will decrease as the cell shape becomes more concave and/or irregular.

## Gene-level QC characteristics

• Proportion of cells expressing each gene

**[B] Segmented cell expression purity**. We implemented two broad classes of statistics to capture (i) the concordance of expression profile with scRNA-seq data and (ii) the expression purity or homogeneity of cell type markers. The scRNA-seq data used are described in Section Datasets and preprocessing and listed in Table 1.

• Concordance with scRNA-seq data - We calculated the similarity of the expression pattern between the segmented cells and publicly available single-cell datasets. Here the similarity was measured by Pearson correlation of the average log-normalised gene expression for each cell type. We also calculated the concordance of the proportion of non-zero expression for each cell type between the segmented cells and scRNA-seq data. For data with paired Chromium data from the same experiment, i.e., Xenium-Brain, we also compared the cell type proportion and quantify the concordance using the Pearson correlation. We annotated the cell type annotation for segmented cells using scClassify[36] with scRNA-seq data as reference.

• Purity of expression - We first curated a list of positive markers and negative markers from the scRNA-seq reference data. For each cell type, we selected the highest and lowest 10 percentile of the genes with difference of expression compared to other cell types. We also removed the positive markers that were common to more than 25% of cell types for a more pure positive marker list. For each segmented cell, we then consider the genes with the highest 10 percentile of expression as positive genes and lowest 10 percentile as negative markers. We then calculated the Precision, Recall and F1 score for both positive and negative markers. We further summarised the average positive marker F1 scores and negative marker F1 scores into one Purity F1 score for each method, where we first scaled the average positive and negative marker F1 scores into the range of [0,1] and then calculated the F1 score of transformed metrics as the following:

$$F1_{\mathrm{purity}} = 2 \cdot \frac{(1 - F1_{\mathrm{negative}}) \cdot F1_{\mathrm{positive}}}{1 - F1_{\mathrm{negative}} + F1_{\mathrm{positive}}}. \qquad (22)$$

## [C] Spatial characteristics

In this category, we measured the association between cell type diversity in local spatial regions and all the cell-level baseline characteristics provided in [A]. We first divided each image into multiple small regions. Then, for each local spatial region, we calculated the cell type diversity using Shannon entropy with the R package 'entropy', where a higher entropy indicates a more diverse cell type composition. Next, we assessed the variability of cell-level baseline characteristics within each local region using the coefficient of variation. Subsequently, for each of the cell-level baseline characteristics mentioned in [A], we calculated the Pearson correlation between the cell type diversity (measured using Shannon entropy) and the coefficient of variation of these characteristics across all local regions. Here, we anticipate that regions with more diverse cell type compositions will exhibit higher variability in cell-level characteristics, leading to a stronger correlation between these two metrics.

## [D] Neighbouring contamination

This metric is designed for cell segmentation to ensure that the expression signals between neighboring cells are not contaminated. For a pair of cell types (e.g., cell type A and B), we computed the Euclidean distance from each cell in cell type A to its nearest neighbor belonging to cell type B. We then grouped the cells of cell type A based on a range of distances. Within each group, we calculated the proportion of cells expressing a selected negative marker, which is a cell type marker for cell type B. We anticipate that the method with less contamination will result in segmented cells expressing lower levels of the negative marker, even when the distance to a different cell type is minimal.

### [E] Replicability

Our analysis involved assessing the agreement between the Xenium-BreastCancer1 and Xenium-BreastCancer2 datasets, which are closely related in terms of all the cell-level baseline characteristics provided in [A]. As these datasets are considered to be sister regions, we anticipated that the distribution of all the baseline characteristics, as well as the cell type composition, would be similar. We use Pearson correlation to quantify the degree of concordance.

### Statistics and reproducibility

All analysis was done in R version version (4.3.0). No statistical method was used to predetermine sample size. No data were excluded from the analyses. All cells that passed quality control were included in the analyses. The experiments were not randomized. The Investigators were not blinded to allocation during experiments and outcome assessment.

### Reporting summary

Further information on research design is available in the Nature Portfolio Reporting Summary linked to this article.

## Data availability

All datasets used in this study are publicly available and were downloaded from the following links (more details including accession codes are provided in Table 1). 10x Genomics Xenium breast cancer replicates 1 and 2: https://www.10xgenomics.com/products/xenium-in-situ/preview-dataset-human-breast. 10x Genomics Xenium mouse brain: https://www.10xgenomics.com/resources/datasets/fresh-frozen-mouse-brain-replicates-1-standard. NanoString CosMx NSCLC: https://nanostring.com/products/cosmx-spatial-molecular-imager/nsclc-ffpe-dataset/. Vizgen MERSCOPE melanoma2: https://info.vizgen.com/merscope-ffpe-solution (requires filling in the form to access). The Stereo-seq E12.5_E1S3 data were downloaded from https://db.cngb.org/stomics/mosta/download/. Tumor Immune Single Cell Hub 2 (TISCH2) BRCA: http://tisch.comp-genomics.org/gallery/?cancer=BRCA&species=Human. 10x Chromium breast cancer: https://www.10xgenomics.com/products/xenium-in-situ/preview-dataset-human-breast. Allen Brain Map Mouse Whole Cortex and Hippocampus SMART-seq: https://portal.brain-map.org/atlases-and-data/rnaseq/mouse-whole-cortex-and-hippocampus-smart-seq. Human Lung Cell Atlas: https://beta.fastgenomics.org/p/hlca. TISCH-NSCLC: http://tisch.comp-genomics.org/gallery/?cancer=NSCLC&species=Human. TISCH-SKCM: http://tisch.comp-genomics.org/gallery/?cancer=SKCM&species=Human. The mouse embryo reference was downloaded from GEO database under accession code [GSE119945]. The TISCH-BRCA datasets were downloaded from GEO database under accession codes [GSE110686], [GSE114727], [GSE138536], [GSE143423], [GSE176078], [GSE148673], [GSE150660]; from EBI database under accession code [E-MTAB-8107]; and from SRA under accession code [SRP114962]. The original published datasets of HLCA can be accessed under GEO accession number [GSE135893] for Banovich_Kropski_2020; URL [https://www.synapse.org/#!Synapse:syn21041850] for Krasnow_2020; [GSE128033] for Lafyatis_Rojas_2019; URL [https://explore.data.humancellatlas.org/projects/c4077b3c-5c98-4d26-a614-246d12c2e5d7] for Meyer_2019; [GSE158127] for Misharin_2021; [GSE122960] and [GSE121611] for Misharin_Budinger_2018; European Genome-phenome Archive study ID [EGAD00001005065] for Teichmann_Meyer_2019. The TISCH-NSCLC datasets were downloaded from GEO database under accession codes [GSE117570], [GSE127465], [GSE143423], [GSE148071], [GSE150660]; and from EBI database under accession code [E-MTAB-6149]. The SKCM datasets were downloaded from GEO database under accession codes [GSE115978], [GSE120575], [GSE123139], [GSE139249], [GSE148190], [GSE72056], [GSE134388], [GSE159251], [GSE166181], and [GSE179373]. Source data are provided with this paper.

## Code availability

We provide our code for data pre-processing, BIDCell training and inference in https://github.com/SydneyBioX/BIDCell, https://doi.org/10.5281/zenodo.10070794[37]. We provide our CellSPA framework in https://github.com/SydneyBioX/CellSPA, https://doi.org/10.5281/zenodo.10295991[38].

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

## Acknowledgements

The authors thank all their colleagues, particularly at the Sydney Precision Data Science Centre and Charles Perkins Centre for their support and intellectual engagement. Special thanks to Yue Cao, Lijia Yu, Andy Tran, and Bárbara Zita Peters Couto for their contributions in weekly discussions, and to Nick Robertson for his contribution to the BIDCell package. Thanks also go to Brett Kennedy and Daniel Dlugolenski from the 10x Genomics team in Australia for providing the initial motivation in discussions.

This work is supported by the AIR@innoHK programme of the Innovation and Technology Commission of Hong Kong to J.Y.H.Y., J.K., E.P., X.F., Y.L. The work is also supported by Judith and David Coffey funding to J.Y.H.Y. and Y.L.; NHMRC Investigator APP2017023 to J.Y.H.Y. and D.M. Australian Research Council Discovery project (DP200103748) to J.K.; Discovery Early Career Researcher Awards (DE220100964) to S.G. and (DE200100944) to E.P. Research Training Program Tuition Fee Offset and Stipend Scholarship to F.A.; Chan Zuckerberg Initiative Single Cell Biology Data Insights grant (2022-249319) to S.G.; and USyd-Cornell Partnership Collaboration Awards to S.G. and D.L. The funding source had no role in the study design, in the collection, analysis, and interpretation of data, in the writing of the manuscript, or in the decision to submit the manuscript for publication.

## Author contributions

J.Y.H.Y. conceived and led the study with design input from E.P. and S.G. X.F. led the development of the method with input and guidance from J.K., J.Y.H.Y., E.P., and Y.L. Y.L. led the development and interpretation of the evaluation framework with input from E.P., S.G., J.Y.H.Y., D.M., and X.F. D.L. performed the data analysis and interpretation of the mouse brain data with input from J.Y.H.Y. and Y.L. Y.L. and X.F. performed all data curation and processing. D.M., C.W., and F.A. contributed to the refinement of the code and evaluation framework with guidance from Y.L. and X.F. All authors contributed to the writing, editing, and approval of the manuscript.

## Competing interests

The authors declare no competing interests.
