## [Peer Review File · Nature Communications]

Biologically-informed self-supervised learning for segmentation of subcellular spatial transcriptomics dataReviewer #1 (Remarks to the Author):

* Summary of the key results

The authors present a deep learning-based approach called BIDCell that uses subcellular spatial transcriptome data, corresponding DAPI images, and average expression profiles of associated single cells for the cell segmentation task. First, I commend the authors for providing an overall clear manuscript, with all graphical presentations very well-regulated. This method is used to learn the relationship between spatially resolved gene expression and cell morphology through multiple constraint methods. However, this work has several major limitations: 1) regarding the comprehensive evaluation of benchmarks, the comments in this section are rather vague 2) the systematic performance evaluation of the model is lacking; 3) the code is poorly readable and difficult to operate. I mainly comment on the following aspects, including result comments, method comments, writing comments, and code reproduction comments. As I describe below, some of these evidences can be strengthened, notably code readability and ease of operation.

The comments provided are below.

**Comments on results

1) Lack of more detailed performance comparison. In the introduction part, the author introduces the cell segmentation work of three conceptual categories (lines 55-91), but there are many methods that have not been compared in the benchmark part, including MESMER, StereoCell, pciSeq, Sparcle, ClusterMap, GeneSegNet, SCS, etc. If these methods are not applicable to the Xenium dataset, the author needs to explain in detail in the method section, explaining why it is not applicable, otherwise the author needs to provide a detailed comparison. Also, in lines 316-323 of the Materials and methods section, the author provides time overhead, but it is only a narrative review that lacks objectivity. The authors should compare the time overhead of all methods, which requires at least a histogram display.

2) BIDCell lacks the application of BGI Stereo-seq. In Line 45, the subcellular transcriptome technology of BGI Stereo-seq is introduced, but the results of this technology are not shown in Lines 243-260. This technology is widely used in embryonic development[1, 2], brain science[3], tumor[4] and other research work, and there is a large amount of data available, so it should be applied to the BIDCell tool. The results of this part can increase the scalability of BIDCell.

3) Model evaluation is vague and lacks objectivity. First of all, a new evaluation system (CellSPA) was established, which is worthy of recognition, but the performance indicators and comments in many places in the article do not correspond. In lines 190-198, the authors describe that "the cell morphology of BIDCell is highly correlated with the nucleus morphology", but the result of Cellpose seems to be more dominant in Fig. 2e. In addition, in Supplementary Figures 5 and 7, the evaluation results of cell morphology did not show that BIDCell has obvious advantages. In lines 208-216, the authors describe BIDCell in terms of high correlation of nuclei and large cell bodies (Fig. 3a), mean percentage of expressed genes (Fig. 3b) and expression purity of segmented cells (Fig. 3d and Supplementary Fig. 8) A significant improvement was achieved, but the result graph shows that the results of Cellpose seem to be better, especially in terms of expression purity in Supplementary Figure 8b. Figure 4c also seems to have this problem. It seems that the results of cellpose nuclei are better, and the prediction results of Cellpose nuclei should be displayed in Fig4 b and e. These results are puzzling to me, please further explanation from the authors.

4) There is a problem with the results section statement. First, the authors describe cell-cell interactions in many places, such as lines 35, 201, 241, 257 and 363, but there is no display of the results of this part. This conclusion at lines 237-241 indicates that it is unreasonable. Cell interaction can include intercellular signal transmission, intercellular adhesion, intercellular immune response, etc. It is not comprehensive to only consider adjacent cells. If the authors wanted to elucidate the results in this regard, they would at least use the Cell Interaction Tool to identify interacting cells, map to spatial locations and then compare with the results predicted by BIDCell. Secondly, Lines 262, BIDCell reveals specific subtypes between neurons, but only some region-specific genes are displayed in lines 276-282, but no subtypes are found. It is recommended that the authors change the wording of these two sections.

**Comment on method

1) Lines 149-151, the author explains that UNnet3+ serves as the backbone framework of BIDCell, and emphasizes that this is not necessary, and can be replaced by this framework, but the rest of the framework is not provided in the code. If the author wants to emphasize the flexibility of the

framework, the author can embed multiple backbone frameworks and let users choose independently. At this time, it is necessary to perform performance evaluation on multiple frameworks, which may require a lot of time overhead. Otherwise, I suggest not emphasizing that the model framework is flexible.

2) The model of BIDCell is novel in design, including input of multiple data types. Compared with other algorithms, BIDCell innovatively uses the corresponding single-cell data to constrain the model, but the overall performance evaluation does not show that there is a difference whether using single-cell data or not. I hope the author will add this part of the discussion.

**Comment on writing

1) The gene name is indicated in italics, and it is recommended that the author make changes throughout the text; for example, lines 276, 280, 281.

2) There are problems in the formula writing of Lines 384, 388, 395, and 400 in the materials and methods part, and should not be written as descriptive formulas. Please refer to the above formula to write.

3) Lines 326-334 in the Materials and methods section, plus references to Supplementary Fig 3.

4) Lines 276-278, the figure that should be cited is Fig 5f.

**Comments on implementation and example code

I downloaded, installed and ran the sample code. But it didn't quite work.

1) Python packages should use standard Python packaging tools. Even better if you use `setuptools` or `poetry` or similar.

2) The demo of the algorithm is not detailed enough, such as processing other sub-resolution platforms except Xenium (BGI Stereo-seq, NanoString CosMx and Vizgen MERSCOPE), and lacks a detailed explanation of the model input in the preprocess step.

3) An error is reported when running the sample code. As a result, an error was reported in "python train.py --config_file configs/config.json" in the "Running BIDCell" step, and some results of the article could not be reproduced. The error code is as follows:

```
"ret = torch._C._nn.nll_loss2d(input, target, weight, _Reduction.get_enum(reduction),
ignore_index)
IndexError: Target -9223372036854775808 is out of bounds. "
```

**References

1. Liu, C., et al., Spatiotemporal mapping of gene expression landscapes and developmental trajectories during zebrafish embryogenesis. *Developmental Cell*, 2022. 57(10): p. 1284-1298. e5.
2. Wang, M., et al., High-resolution 3D spatiotemporal transcriptomic maps of developing *Drosophila* embryos and larvae. *Developmental Cell*, 2022. 57(10): p. 1271-1283. e4.
3. Wei, X., et al., Single-cell Stereo-seq reveals induced progenitor cells involved in axolotl brain regeneration. *Science*, 2022. 377(6610): p. eabp9444.
4. Wu, L., et al., An invasive zone in human liver cancer identified by Stereo-seq promotes hepatocyte-tumor cell crosstalk, local immunosuppression and tumor progression. *Cell Research*, 2023: p. 1-19.

Reviewer #2 (Remarks to the Author):

Fu et al. present BIDCell, a method for subcellular spatial transcriptomics (SST) cell segmentation. They clearly describe the existing literature and the methodological improvements introduced in BIDCell. Namely, BIDCell incorporates what they term "biologically-informed loss functions" and prior knowledge from existing scRNA-seq data. They also introduce an evaluation framework for cell segmentation, CellSPA.

Overall, I believe the methodological advances of BIDCell are useful, particularly the use of scRNA-seq as prior information. It seems logical that the introduced components of the loss function would help performance and increase the interpretability of the model as compared to e.g. Cellpose, which is fully black box. However, I do have a few major concerns about the evaluation

of the method, especially as compared to Cellpose (it seems clear that BIDCell and Cellpose are generally outperforming the others).

Major concerns

1) Based on Figures 3a,b,d, 2e, and 4c, does it seem like Cellpose is doing better? In results section 3, the text mentions the tradeoff between average total transcript per cell and level of correlation, but couldn't the argument be made that Cellpose has higher specificity of transcript assignment? Given these figures, the authors should at least relax claims of being the "best" in the Results section and talk about the advantages and disadvantages of both methods more in the Discussion.

2) I think the contamination plots in Figure 3e / Supplementary Figure 9 have the potential to be the most convincing argument for why BIDCell would be better, and would make sense given the use of scRNA-seq as a prior, but at the moment they are a bit confusing.

- For example, for "CD8B in B Cells" - why do many of the lines increase going from 20,30 - 30,40?

- Does the amount of transcript being expressed also matter? Or is it very sparse?

- To address these concerns, I think it would be most useful to have a figure that captures the variability better, e.g. a boxplot, where method is on the x-axis, and y-axis is authors' choice of contamination metric, including results combining all the marker contaminations.

3) While the use of "biologically-informed loss functions" is potentially interesting, I don't see a thorough enough discussion of how each of the components is useful. They certainly make sense biologically, but the relative contribution of each component is not clear.

- The authors could add in more text about the ablation study and its biological interpretation (Supp. Fig. 3). For example, it seems interesting that not including the positive markers loss function("loss_pos", I assume) performs the worst.

- Including the Cellpose dot in the ablation experiment in Supp. Fig. 3 panels c-e would help with comparing the methods.

- Did the authors test other values besides $\lambda=1$ (other than the ablation experiment)? Would cross validation allow these lambdas to be tuned according to each data set and perhaps result in better performance?

Minor points

1) It seems to me that for cell segmentation, people care mostly about a) the cell typing/purity of transcript assignments and b) the visual look of the segmentation. I'm not sure that the rest of the metrics introduced in CellSPA are particularly useful. In particular, the authors should at least mention what conclusions (if any) we can draw from looking at the panels of Figure 2. For example, number of transcripts per cell - is more necessarily better? Couldn't some cell types have less transcripts per cell?

Should the phrase "% of cells expressed per gene" be "% of cells expressing each gene"?

2) The ablation study is referred to in Results Section 1, Supplementary Figure 3, but not explicitly mentioned as "ablation study" in the main text until line 244 ("As an additional sensitivity analysis to the ablation study")

3) Figure 1 - In "Deep learning model" - the abbreviations E1-E5, D1-D4 should be explained in the text or figure legend, in addition to the dotted lines and the meaning of the colors of the dotted lines.

4) Figure 2 - should say that 'p' in the elongation figure is correlation or change it to r or cor in line with other areas of the text, otherwise it's easy to confuse with p-value.

Methods

- Apologies if I missed this somewhere, but is it "self-supervised" because you use a separate procedure to generate labels prior to fitting the deep learning model?

- How are overlapping patches aggregated if there is disagreement in the overlapping regions?

Data and source code

It seems the data and source code is available on the public BIDCell GitHub repository. Currently, the authors' code is in the form of a collection of standalone python scripts that are run in succession to preprocess, train, and postprocess. I think this is OK but ideally could be converted into a package form rather than sequentially running python scripts. In its current form, it would probably take some time to edit these steps for another data set, but this is probably unavoidable given the complex nature of the segmentation problem.

In addition, following the instructions as written on the main GitHub README, I ran the steps without changing anything and encountered the error "IndexError: Target - 9223372036854775808 is out of bounds" when running the python train.py -config_file configs/config.json. Not sure if the authors have seen this before.

Reviewer #3 & 4 (Remarks to the Author):

The authors propose a novel methodology to refine image-based spatial transcriptomics cell segmentation called biologically-informed deep learning-based cell segmentation (BIDCell) that takes into account the molecular data. They also present an evaluation suit: Cell Segmentation Performance Assessment (CellSPA) which compares performance measured with a variety of metrics. They compare different methods extensively using publicly available data acquired with different technologies.

The paper provides extremely interesting metrics for the very difficult problem of evaluating segmentation when ground-truth data is unavailable, for example, evaluating the segmentation quality given positive and negative cell marker gene expression, comparing the tradeoff between transcripts per cell, and correlation with scRNA-seq results. These metrics are implemented on the companion software CellSPA.

In our opinion, a major point to improve is the following: we had difficulty appreciating the value of some of the metrics. We suggest that the proposed evaluation benchmark should be improved by providing more extensive details of how each metric benefits the estimation of the similarity between the segments and the true unknown correct cell segments.

The remaining comments are itemized below so they can be objectively answered and quoted.

Comments:

1. Why is the method "self-supervised" if it uses negative and positive markers on the training loss? Wouldn't "weakly-supervised" be more appropriate?
2. Segments should be overlaid over nuclei and cytoplasm (or membrane) channels, H&E could be used if that's not available because it's harder to evaluate the segmentation masks over transcripts.
3. Could you elaborate on how some points are below the red diagonal line in Supplementary Figure 4? From my understanding, a "segmented cell" is a superset of a "nucleus". Therefore, it should always contain more transcripts. Is it because these methods do not take the initial "nucleus" segmentation as a hard constraint?
4. Data from CellSPA data should be stored in permanent and versioned storage and not google drive. Is there a difference with the data of the 10x resource?
5. The authors could spend more time explaining how each metric contributes to better evaluating the segmentation performance. Otherwise, it's not clear how each individual metric benefits the CellSPA package.
6. Material & methods L186: Why are the background probabilities averaged? Won't this overestimate the background since each cell's instance probability prediction would classify other cells' instances as background?
7. Which lambda (without subscript) is referred to in L289 of material & methods? Is it all of them? It could be more clear.

8. It has been shown that torch Adam weight-decay is not L2 regularization [1]. Therefore, equation 13 from material & methods should be corrected.
9. Why was the watershed baseline (L363 from material & methods) run on the sum of the molecule channels and not the cytoplasm or membrane channel? The sum of molecule channels does not yield a topology that is adequate for the watershed operator, which splits segments on the high-gradient features.
10. Why are Circularity (L395 from material & methods) and Compactness (L406) needed when they are extremely similar?
11. How much does the morphological post-processing (L192 from material & methods) impact the proposed method results? Several metrics evaluate the convexity and smoothness of the contours in some way, which are improved by the morphological post-processing. Wouldn't this post-processing also benefit the other methods?
12. Why is the entropy compared with the coefficient of variation (CV) using Pearson correlation? Is there any reason why this relationship should be linear? Would a rank-based correlation be more appropriate since the CV depends on the scale of the distribution?
13. Could the authors provide more details on the Neighbouring Contamination metric? A figure would really assist in understanding it.

Minor comments:

1. Use the same ggplot2 theme as the rest of the article in Supplementary Figure 1.
2. Supplementary Figure 3 could be improved by switching from sequential to high-contrast colors.
3. It would greatly improve the article to have the plots represented as vector graphics, especially because several plots need to be zoomed in. For example, Figures 1,2,3, etc.
4. Should it be "p = ...", "R = ..." or "cor = ..." on Figure 2e? It would be helpful to provide more information on what this measurement means, besides correlation, because there are several methods of correlation.
5. Typo on figures' legend, Fibroblast -> Fibroblast
6. Figure 3a correlation should be normalized between 0 and 1.0 to avoid misleading an unattentive reader.
7. L338 material & methods, Table 1 should be Table 2.
8. L431 material & methods, Table 2 should be Table 1.
9. Shouldn't the Supplementary Figure 8 axis be on the same scale between different marker genes? If not, why not?
10. How significant is the difference between 1 - Neg F1 and Pos F1 scores from Supplementary Figures 12d and 13d? They are all on the second decimal level.

References:

- [1] Loshchilov, Ilya, and Frank Hutter. "Fixing weight decay regularization in Adam." (2018).

REVIEWER COMMENTS

Reviewer #1 (Remarks to the Author):

* Summary of the key results

The authors present a deep learning-based approach called BIDCell that uses subcellular spatial transcriptome data, corresponding DAPI images, and average expression profiles of associated single cells for the cell segmentation task. First, I commend the authors for providing an overall clear manuscript, with all graphical presentations very well-regulated. This method is used to learn the relationship between spatially resolved gene expression and cell morphology through multiple constraint methods. However, this work has several major limitations: 1) regarding the comprehensive evaluation of benchmarks, the comments in this section are rather vague 2) the systematic performance evaluation of the model is lacking; 3) the code is poorly readable and difficult to operate.

Response: We thank the reviewer for the positive feedback and the suggestions. We have addressed each of the limitations raised as part of the detailed comments provided below. In particular, for the limitations mentioned above:

- (1) For evaluation, we have added additional documentation surrounding our comprehensive evaluation and in particular, addressed the issues surrounding:
 - Inclusion and exclusion reasons for related methods (page 1);
 - Application to additional data, including those of Stereo-seq (page 3).
- (2) For performance evaluation, CellSPA provides a comprehensive and systematic performance of the full approach that is beyond a typical ablation study. We have now further extended the ablation study as detailed in the response on page 10.
- (3) We have now created a Python package for BIDCell, with special attention placed on:
 - Prerequisites or dependencies and software versioning;
 - Installation steps and instructions;
 - Usability and readability;
 - Examples to provide basic usage instructions for a demo dataset and various SST platforms.

I mainly comment on the following aspects, including result comments, method comments, writing comments, and code reproduction comments. As I describe below, some of these evidences can be strengthened, notably code readability and ease of operation.

The comments provided are below.

**Comments on results

1) Lack of more detailed performance comparison. In the introduction part, the author introduces the cell segmentation work of three conceptual categories (lines 55-91), but there are many methods that have not been compared in the benchmark part, including MESMER, StereoCell, pciSeq, Sparcle, ClusterMap, GeneSegNet, SCS, etc. If these methods are not applicable to the Xenium dataset, the author needs to explain in detail in the method section, explaining why it is not applicable, otherwise the author needs to provide a detailed comparison.

Response: We thank the reviewer for the suggestions. We briefly mentioned these other methods in the Introduction to provide an overview of previous related efforts. However, as alluded to by the reviewer, not all of these methods are suitable for performance comparison. We have added in Supplementary Table 1 more details on reasons for exclusion; and revised our manuscript to better describe our inclusion criteria for comparison methods in the Performance Evaluation section in Materials and Methods:

“Methods that were excluded from the evaluations include those that focus on the assignment of transcripts to cells and do not consider the cell boundaries, underperformance on the public datasets, lack of code and instructions to prepare data into the required formats, and failure of the method to detect any cells (Supplementary Table 1).”

We also include representative examples below (Figures R1 and R2) that show the segmentations predicted by Mesmer and StereoCell, which were consistently rounded morphologies that do not demonstrate the diverse morphologies expected for the cell types. Please note that our aim is not to provide a comprehensive benchmarking of all related methods in this manuscript, but to compare the most representative methods to BIDCell from different approaches.

Figure R1: Example segmentations from various methods including Mesmer and StereoCell for Xenium-BreastCancer1. H&E images are shown for visualisation purposes only.

Figure R2: Example segmentations from various methods including Mesmer and StereoCell for Xenium-MouseBrain.

Also, in lines 316-323 of the Materials and Methods section, the author provides time overhead, but it is only a narrative review that lacks objectivity. The authors should compare the time overhead of all methods, which requires at least a histogram display.

Response: As suggested, we have added the time overhead in Supplementary Figure 26:

Supplementary Figure 26: Bar plot showing a comparison of the runtimes of different methods when applied to Xenium-BreastCancer1 (34 million transcripts). BIDCell achieved superior performance with a reasonable runtime. We note that we used a GPU with 12GB VRAM. BIDCell will run faster on GPUs with more VRAM, which is commonly at least 24GB.

2) BIDCell lacks the application of BGI Stereo-seq. In Line 45, the subcellular transcriptome technology of BGI Stereo-seq is introduced, but the results of this technology are not shown in Lines 243-260. This technology is widely used in embryonic development[1, 2], brain science[3], tumor[4] and other research work, and there is a large amount of data available, so it should be applied to the BIDCell tool. The results of this part can increase the scalability of BIDCell.

Response: We believe that our ability to extend to three technologies will enable the flexibility to extend to a fourth, and accessibility to publicly available data was an important initial consideration on our choice of the technologies. While a number of papers claim the public availability of BGI Stereo-seq data, it is important to note that some repositories require a long application process and may not be accessible due to jurisdictional restrictions.

That being said, we have strengthened the manuscript by following the reviewer's suggestions to apply BIDCell to BGI Stereo-seq data. We select Stereo-seq data that can be publicly downloaded from a website and here, we apply BIDCell to the Stereo-seq mouse embryo dataset from Chen and colleagues published in Cell. (Chen et al. 2022). We have added the results to Supplementary Figure 21 and added details about this dataset and results in our manuscript:

Ref: Chen, Ao, Sha Liao, Mengnan Cheng, Kailong Ma, Liang Wu, Yiwei Lai, Xiaojie Qiu, et al. 2022. "Spatiotemporal Transcriptomic Atlas of Mouse Organogenesis Using DNA Nanoball-Patterned Arrays." *Cell* 185 (10): 1777–92.e21.

Under Results on MS page 6:

“Furthermore, we also applied *BIDCell* to Stereo-seq from BGI (Supplementary Figure 21). We have now demonstrated the applicability of *BIDCell* on data from four major platforms, and from five different tissue types. We believe that our method has the flexibility and generalisability to other data from other SST platforms and tissues.”

Under Materials and Methods on pages 2 and 3:

Stereo-seq-MouseEmbryo. The Stereo-seq data used in this study, including the DAPI image and detected gene expressions (bin 1), were downloaded from <https://db.cngb.org/stomics/mosta/download/> for sample E12.5_E1S3. Stereo-seq data contains a far greater number of genes compared to Xenium, CosMx, and MERSCOPE. For efficiency, we selected a panel of 275 highly variable genes (HVGs) as the input to *BIDCell*. The HVGs are the common genes of the top 1,000 HVGs from both Stereo-seq data and the single-cell reference data.

Mouse Embryo reference. The reference for Stereo-seq-MouseEmbryo was downloaded from GEO database under accession code GSE119945: <https://www.ncbi.nlm.nih.gov/geo/query/acc.cgi?acc=GSE119945> (Cao et al., 2019), which contain both counts and cell type annotation data. The E12.5 data was then used as reference.

Supplementary Figure 21: Application of *BIDCell* to mouse embryo data from BGI Stereo-seq. (a) Example cell segmentations from *BIDCell* and *StereoCell*; and (b) examples of marker expression of spatial region for Chen et al. (Bin 50), *BIDCell*, and *StereoCell*.

3) Model evaluation is vague and lacks objectivity. First of all, a new evaluation system (CellSPA) was established, which is worthy of recognition, but the performance indicators and comments in many places in the article do not correspond.

Response: We appreciate the reviewer's recognition of the significant contribution of CellSPA and their thoughtful suggestions to enhance the clarity in objectivity of our manuscript. To increase clarity for our proposed evaluation scores or metrics is the need to interpret them relative to the "size of the segmented cells" and not a direct ranking of the "scores". We have added further explanations throughout the manuscript, as highlighted below.

In lines 190-198, the authors describe that "the cell morphology of BIDCell is highly correlated with the nucleus morphology", but the result of Cellpose seems to be more dominant in Fig. 2e.

Response: Our initial statement (line 190) about "correlation of cell morphology between cell body and nuclei", suggests that cell body segmentation from BIDCell ($r = 0.95$) has very high similarity. Hence in Figure 2e, we are illustrating our similarity with nuclei-based methods and not competing with nuclei-based methods.

Next, when compared to all other methods that generate similar sizes of cells with BIDCell, we found that our method achieves a significantly higher correlation (others range from 0.2 to 0.8). It is important to note that Cellpose has an average cell size similar to the nuclei-based approach and much smaller than the average cell size of BIDCell and other approaches. We now make a clearer statement and added an additional figure that highlights the need to interpret the morphology metric relative to cell size (Figure 2f).

In addition, in Supplementary Figures 5 and 7, the evaluation results of cell morphology did not show that BIDCell has obvious advantages.

Response: It is important to note that the point we aim to deliver through Supplementary Figures 5 and 7 (now 10 and 12) is that the segmented cells from BIDCell exhibit more diverse cell morphology characteristics for cells from different cell types compared to other methods, as it is expected that cells can have different cell shapes, areas and morphology, therefore the cell morphology statistics should have a high variability. In contrast, other methods output more homogeneous cell morphologies (MS lines 197-199).

In lines 208-216, the authors describe BIDCell in terms of high correlation of nuclei and large cell bodies (Fig. 3a), mean percentage of expressed genes (Fig. 3b) and expression purity of segmented cells (Fig. 3d and Supplementary Fig. 8) A significant improvement was achieved, but the result graph shows that the results of Cellpose seem to be better, especially in terms of expression purity in Supplementary Figure 8b. Figure 4c also seems to have this problem. It seems that the results of cellpose nuclei are better, and the prediction results of Cellpose nuclei should be displayed in Fig4 b and e. These results are puzzling to me, please further explanation from the authors.

Response: It is important to note that, on one hand, a direct comparison to Cellpose nuclei or any approach that focuses only on the segmentation of nuclei is not appropriate because we are trying to identify the cell body (nuclei plus other parts of the cells). As noted at the start of our response, purity alone is not an appropriate measure, as one can achieve high purity by only including nuclei or even dividing nuclei as a "segmented cell". The smaller "segmented cells" tend to have higher purity because they are closer to only include transcripts that are near nuclei (or even divide nuclei into parts) and the nuclei usually contain the main signals of the cell identity. Our interest is to identify the

cell body, that is, a cell segmentation method should not only include signals about nuclei, but also extend the cell boundary to include signals from cell cytoplasm. With a large cell size, the purity often decreases as it inevitably includes noise.

We consider that a good cell body segmentation to include

- (i) signals from nuclei and other part of cells; and
- (ii) at the same time maintain the purity of the cells.

That is we aim to assess methods that can achieve a balance between cell body segmentation and expression purity. That being said, we have now included

- the Cellpose nuclei results in Figure 4b and 4e (as suggested);
- clarified this concept in the discussion (MS page 8).
- illustrated this joint assessment in Figure 3a, b and d and Figure 4c.

Note, although Cellpose usually has better correlation and purity in expression related metrics, it segments a much smaller cell area and assigns less proportion of transcripts compared to BIDCell. We demonstrate that BIDCell reaches the ideal balance between these two aspects of cell segmentation performance.

Updated Figure 4b: Comparative illustration of the predictions from BIDCell, NanoString and Cellpose nuclei for CosMx-Lung.

Updated Figure 4e: Comparative illustration of the predictions from BIDCell, NanoString and Cellpose nuclei for MERSCOPE-Melanoma.

4) There is a problem with the results section statement. First, the authors describe cell-cell interactions in many places, such as lines 35, 201, 241, 257 and 363, but there is no display of the results of this part. This conclusion at lines 237-241 indicates that it is unreasonable. Cell interaction can include intercellular signal transmission, intercellular adhesion, intercellular immune response, etc. It is not comprehensive to only consider adjacent cells. If the authors wanted to elucidate the results in this regard, they would at least use the Cell Interaction Tool to identify interacting cells, map to spatial locations and then compare with the results predicted by BIDCell.

Response: We thank the reviewer for the insightful comments and acknowledge that there are multiple categories of cell cell interaction (CCI). We have reworded the manuscript throughout to reduce the focus on CCI. In particular, we reworded the title of our second result section as

“BIDCell captures improved purity of cell expression, leading to less contamination from neighbouring cells.”

Furthermore, we illustrate here how accurate cell segmentation allows us to better examine CCI between cell types with the spatial proximity and physical interaction constraint. Following the reviewer’s suggestion, we use a cell interaction tool (NeuronChat, (Zhao et al. 2023)) to identify interacting cells that can be mapped to spatial locations to illustrate the utility of BIDCell. We performed NeuronChat on the mouse brain data generated by Xenium as an example.

Methods:

- (1) We estimate the ligand receptor pairs using scRNA-seq data because only ~12% of ligand receptor genes of the database are captured in the SST data. Following the NeuronChat tutorial (Zhao et al.), we used the SMART-seq2 data from Allen Brain Map to first estimate the

intercellular interaction network and then filter the network by cell proximity information estimated from the segmented Xenium data.

(2) We focus on the cell type interaction network in the hippocampus region (Figure 5) constructed using NeuronChat under two settings and shown in Figure R3:

- (a) without any spatial proximity constraint implemented in NeuronChat (left panel);
- (b) filtering strategy which focuses only on physical interactions, using our BIDCell segmentation results (right panel).

Results:

The Figure R3 below illustrates the cell type interaction network derived from NeuronChat using two different settings. We can see that without constraint, the left panel shows a close to fully connected network, potentially with a large amount of false positives, while the right panel allows us to filter the signals and focus on examining the cell type interaction network under physical contact constraints. In addition, it is important to note that the filtering method (right panel) is only possible with segmented cell (SST images) and not possible for “spot-based” results or without accurate cell morphology estimation.

Reference

Zhao, Wei, Kevin G. Johnston, Honglei Ren, Xiangmin Xu, and Qing Nie. 2023. “Inferring Neuron-Neuron Communications from Single-Cell Transcriptomics through NeuronChat.” *Nature Communications* 14 (1): 1128.

Figure R3: The figure shows the NeuronChat interaction network based on the cell type interaction network in the hippocampus region (Figure 5) constructed using NeuronChat under three settings without spatial constraint (left panel) and with physical contact constraints (right panel).

Secondly, Lines 262, BIDCell reveals specific subtypes between neurons, but only some region-specific genes are displayed in lines 276-282, but no subtypes are found. It is recommended that the authors change the wording of these two sections.

Response: Given the spatial context of our data, our main challenge is to identify markers for sub-regions. Segmenting cells with BIDCell enables neuronal subtypes to be distinguished more clearly using specific markers. We have added two sentences and new literature to illustrate this point in the Results section (MS page 7).

“BIDCell can identify neuronal subtype markers that distinguish granule neurons in the dentate gyrus (Prox1) from pyramidal neurons in CA1-3 (Neurod6) ((Hamilton et al. 2017); Figure 5f). Furthermore,

it is able to spatially subdivide pyramidal neurons in the CA region despite their close proximity to one another. Figure 5f shows the expression patterns of Wfs1 in CA1 (Dong et al., 2009), Necab2 in CA2 (Zimmermann et al., 2013) and Slit2 in CA3 (Blockus et al., 2021), consistent with prior studies. Interestingly, we found a new gene (Cpne8) that is enriched in CA1, consistent with in situ data from the Allen Brain Atlas and illustrates BIDCell's capacity for biological discovery."

Ref: Hamilton, D. J., C. M. White, C. L. Rees, D. W. Wheeler, and G. A. Ascoli. 2017. "Molecular Fingerprinting of Principal Neurons in the Rodent Hippocampus: A Neuroinformatics Approach." *Journal of Pharmaceutical and Biomedical Analysis* 144 (September): 269–78.

****Comment on method**

1) Lines 149-151, the author explains that UNet3+ serves as the backbone framework of BIDCell, and emphasizes that this is not necessary, and can be replaced by this framework, but the rest of the framework is not provided in the code. If the author wants to emphasize the flexibility of the framework, the author can embed multiple backbone frameworks and let users choose independently. At this time, it is necessary to perform performance evaluation on multiple frameworks, which may require a lot of time overhead. Otherwise, I suggest not emphasizing that the model framework is flexible.

Response: We thank the reviewer for the suggestions. As the reviewer has noted, the BIDCell framework with its loss functions are designed to allow flexibility with the choice of architecture of the deep learning backbone that maps transcripts to cell morphology (i.e., segmentations). The backbone is one component of the framework that may be treated in a modular manner and be swapped out with alternatives. We have now added this flexibility into our code on GitHub. The user can either select from a predefined list of popular segmentation backbones such as ResNet and DenseNet, or define their own custom backbone in the `SegmentationModel` class in `model.py`.

The default selection is still UNet3+, and we have shown that it performs well across a range of technologies and tissue types. Furthermore, we have added segmentation results using alternative backbones (U-Net, and U-Net with a ResNet18, DenseNet121, or EfficientNet encoder) in Supplementary Figure 7, demonstrating the flexibility and stability of our BIDCell framework with alternative backbones.

Supplementary Figure 7: Performance of BIDCell with different backbones (U-Net, U-Net-ResNet18, U-Net-DenseNet121, and U-Net-EfficientNetB1). These results further demonstrate the flexibility of our BIDCell framework with alternative backbones, as it achieved consistently high performance, with relatively high purity and additional transcript information to the nuclei. (a) Box plots of the number of transcripts and genes per cell for the different settings; (b) scatter plot showing genes expressed in the segmented cells compared to the nuclei; (c) scatter plot between Pearson correlation with Chromium (y-axis) and total transcripts per cell (x-axis); (d) scatter plot between inverse negative marker F1 (y-axis) and positive marker F1 (x-axis); and (e) scatter plot between purity F1 (y-axis) and total transcripts per cell (x-axis).

2) The model of BIDCell is novel in design, including input of multiple data types. Compared with other algorithms, BIDCell innovatively uses the corresponding single-cell data to constrain the model, but the overall performance evaluation does not show that there is a difference whether using single-cell data or not. I hope the author will add this part of the discussion.

Response: We thank the reviewer for the suggestions. BIDCell uses single-cell data to inform the positive and negative loss functions, and enables the model to predict elongated cell shapes. We have added experiments in the ablation study: when both marker losses are removed during training, and when the ability to predict elongated cells is removed (`loss_posneg` and `no_elongate`). We have added this in Supplementary Figure 4 and the additional results to the revised manuscript in the Results section (MS page 4).

“With the inclusion of single-cell data (which informs the positive and negative losses, and contributes to the ability to predict elongated cell shapes), performance improved considerably, particularly in purity metrics and correlation to Chromium data. The use of single-cell data helped the model to better capture transcripts that are more biologically meaningful within cells.”

Supplementary Figure 4: Ablation study of different settings of main parameters: *ekernel_50* and *ekernel_70*, where l_i of the elliptical kernel in the cell-calling loss was 50 or 70; *kernel_10* and *kernel_30*, where the diameter of the circular kernel in the cell-calling loss was 10 and 30; *loss_**, where each loss was individually set to zero; *no_elongate*, where the ability to predict elongated cells is removed; and *BIDCell*, the proposed settings. (a) Box plots of the number of transcripts and genes per cell for the different settings; (b) scatter plot showing genes expressed in the segmented cells compared to the nuclei; (c) scatter plot between Pearson correlation with Chromium (y-axis) and total transcripts per cell (x-axis); (d) scatter plot between inverse negative marker F1 (y-axis) and positive marker F1 (x-axis); and (e) scatter plot between purity F1 (y-axis) and total transcripts per cell (x-axis). All the loss functions except the cell-calling loss increased the expression purity. The cell-calling loss increased the number of transcripts captured, without detriment to expression purity. The nucleus encapsulation loss improved purity metrics, by leveraging genes expressed in nuclei to guide the capture of genes expressed in cells. The overlap and over-segmentation losses improve the shapes of predicted cells, which translates into improved purity metrics. The differences between *BIDCell* and *loss_posneg* show that the marker losses improved performance considerably, particularly in purity metrics and Pearson correlation to Chromium data, while more transcripts were captured in the segmented cells.

**Comment on writing

1) The gene name is indicated in italics, and it is recommended that the author make changes throughout the text; for example, lines 276, 280, 281.

Response: As suggested, we have indicated the gene names in italics.

2) There are problems in the formula writing of Lines 384, 388, 395, and 400 in the materials and methods part, and should not be written as descriptive formulas. Please refer to the above formula to write.

Response: As suggested, we have changed these formulas for baseline metrics to non-descriptive formulas in the revised manuscript.

3) Lines 326-334 in the Materials and methods section, plus references to Supplementary Fig 3.

Response: As suggested, we have added a reference to the figure in this paragraph.

4) Lines 276-278, the figure that should be cited is Fig 5f.

Response: We have corrected this in the revised manuscript.

****Comments on implementation and example code**

I downloaded, installed and ran the sample code. But it didn't quite work.

1) Python packages should use standard Python packaging tools. Even better if you use `setuptools` or `poetry` or similar.

Response: We thank the reviewer for the suggestions. We agree that the code is ideally a package that a user can easily install and call functions from, rather than having to run individual scripts. We have created a Python package for BIDCell and its associated functions, including data processing. The code may be found at <https://github.com/SydneyBioX/BIDCell>. We have plans to continue to support and extend the code after this manuscript.

2) The demo of the algorithm is not detailed enough, such as processing other sub-resolution platforms except Xenium (BGI Stereo-seq, NanoString CosMx and Vizgen MERSCOPE), and lacks a detailed explanation of the model input in the preprocess step.

Response: We thank the reviewer for the suggestions. We have added new functionality to our code, and it now may be applied to other platforms such as Stereo-seq, CosMx, MERSCOPE, and Xenium. We included instructions in the README, and also example scripts for the entire data processing and segmentation pipeline for different platforms to make the code easier to use.

3) An error is reported when running the sample code. As a result, an error was reported in "python train.py --config_file configs/config.json" in the "Running BIDCell" step, and some results of the article could not be reproduced. The error code is as follows:

```
"ret = torch._C._nn.nll_loss2d(input, target, weight, _Reduction.get_enum(reduction), ignore_index)
IndexError: Target -9223372036854775808 is out of bounds."
```

Response: We thank the reviewer for raising this issue. We have found the cause of this error to be from a bug with the code for the positive and negative marker losses. The bug caused this value -9223372036854775808 (-2^{63}) to be generated, which is an invalid prediction class target. We have fixed this bug and pushed it to our GitHub repository.

Reviewer #2 (Remarks to the Author):

Fu et al. present BIDCell, a method for subcellular spatial transcriptomics (SST) cell segmentation. They clearly describe the existing literature and the methodological improvements introduced in BIDCell. Namely, BIDCell incorporates what they term "biologically-informed loss functions" and prior knowledge from existing scRNA-seq data. They also introduce an evaluation framework for cell segmentation, CellSPA.

Overall, I believe the methodological advances of BIDCell are useful, particularly the use of scRNA-seq as prior information. It seems logical that the introduced components of the loss function would help performance and increase the interpretability of the model as compared to e.g. Cellpose, which is fully black box. However, I do have a few major concerns about the evaluation of the method, especially as compared to Cellpose (it seems clear that BIDCell and Cellpose are generally outperforming the others).

Major concerns

1) Based on Figures 3a,b,d, 2e, and 4c, does it seem like Cellpose is doing better? In results section 3, the text mentions the tradeoff between average total transcript per cell and level of correlation, but couldn't the argument be made that Cellpose has higher specificity of transcript assignment?

Response: We thank the reviewer for this important suggestion. We agree that it is imperative to clarify this trade-off in the performance of cell segmentation methods. We have discussed this point in detail in response to comment 1.3 to clarify the different types of comparisons made. Overall, considering multiple aspects of comparison, Cellpose does not perform better.

Given these figures, the authors should at least relax claims of being the “best” in the Results section and talk about the advantages and disadvantages of both methods more in the Discussion.

Response: To avoid possible over-claims, we have carefully checked our manuscript, and we identified one sentence (Line 202-203) where we employed the term “best” in the context of “best balance between high correlation with segmented nuclei and a large cell body among all methods.” Here “best” refers to the balance between the size of segmented cells and expression profile similarity with the scRNA-seq reference. We have rephrased “best balance” to “ideal balance” to better capture our point.

To clarify this point, we have rewritten the fourth paragraph in the discussion to elaborate more on the trade-off between these two aspects:

“A comprehensive evaluation framework is vital when comparing diverse segmentation approaches in the absence of a ground truth. It is important to recognise that different segmentation approaches may purposefully have different priorities and outcomes. For example, a segmentation approach such as a seeded Voronoi tessellation will identify larger cells than a fixed expansion around the nuclei, such as Cellpose cell. The former will typically assign more transcripts and produce a denser map of which cells are touching. In contrast, the latter may produce more homogenous profiles of the cells with fewer assigned molecules and tighter cell boundaries, limiting its capability to estimate physical cell interactions. While achieving more homogenous cell bodies is desired, it can also result from the arbitrary over-segmentation of nuclei. This emphasises that the use of employing a variety of metrics to quantify segmentation performance enables a systematic assessment and reveals the desirable properties of each approach.”

2) I think the contamination plots in Figure 3e / Supplementary Figure 9 have the potential to be the most convincing argument for why BIDCell would be better, and would make sense given the use of scRNA-seq as a prior, but at the moment they are a bit confusing.

- For example, for “CD8B in B Cells” - why do many of the lines increase going from 20,30 - 30,40?
- Does the amount of transcript being expressed also matter? Or is it very sparse?
- To address these concerns, I think it would be most useful to have a figure that captures the variability better, e.g. a boxplot, where method is on the x-axis, and y-axis is authors' choice of contamination metric, including results combining all the marker contaminations.

Response: Thank you for this comment. To further illustrate the concept of this metric, we have included a schematic of this metric (graphical illustration) in the Supplementary Figure 8. The region 30-40 is a relatively larger distance, and as such we only observed a very small number of cells and thus expect higher variability. In this case, a small number of cells (3~8 cells) expressing this marker in the region (30-40) may show an unexpected upward trend. To account for this, it is important to focus on regions much closer for contamination interpretation. In addition, we agree that the number of transcripts being expressed also matters. As suggested, we explored some other approaches to

evaluation, including using boxplots to visualise the number of transcripts being expressed. But as the reviewer notes, it is hard to distinguish the performance of the methods using boxplot because of the sparsity of the data, as shown in Figure R4 (right panel). Therefore, we decide to use the original designed metric - proportion of cells expressing negative markers.

Figure R4: Boxplots showing the number of cells for a given distance (left panel) and average expression values (right panel).

3) While the use of “biologically-informed loss functions” is potentially interesting, I don’t see a thorough enough discussion of how each of the components is useful. They certainly make sense biologically, but the relative contribution of each component is not clear.

- The authors could add in more text about the ablation study and its biological interpretation (Supp. Fig. 3). For example, it seems interesting that not including the positive markers loss function (“loss_pos”, I assume) performs the worst.

Response: Thank you and we have revised the manuscript to enhance clarity regarding the relative contribution of each component of the loss function. We expanded the ablation study to investigate the contribution of the loss functions, by individually setting each loss to 0 (Supplementary Figure 4). We have added an experiment in the ablation study where both the positive and negative marker losses are set to 0 (`loss_posneg`). We have also added some visual illustrations of the segmentation predictions from different ablation experiments in Supplementary Figure 5. These results help to illustrate the effects of each loss component.

We have added Supplementary Figures 4 and 5, and results to MS page 4.

“We investigated removing individual losses in an ablation study with Xenium-BreastCancer1 data (Supplementary Figures 4 and 5). Our investigation shows that the losses work synergistically; e.g., there was a marked increase in purity F1 relative to the amount of captured transcripts when the losses were combined. With the inclusion of single-cell data (which informs the positive and negative losses, and contributes to the ability to predict elongated cell shapes), performance improved considerably, particularly in purity metrics and correlation to Chromium data. The use of single-cell data helped the model to better capture transcripts that are more biologically meaningful within cells.”

Supplementary Figure 4: Ablation study of different settings of main parameters: `ekernel_50` and `ekernel_70`, where l_i of the elliptical kernel in the cell-calling loss was 50 or 70; `kernel_10` and `kernel_30`, where the diameter of the circular kernel in the cell-calling loss was 10 and 30; `loss_*`, where each loss was individually set to zero; `no_elongate`, where the ability to predict elongated cells is removed; and `BIDCell`, the proposed settings. (a) Box plots of the number of transcripts and genes per cell for the different settings; (b) scatter plot showing genes expressed in the segmented cells compared to the nuclei; (c) scatter plot between Pearson correlation with Chromium (y-axis) and total transcripts per cell (x-axis); (d) scatter plot between inverse negative marker F1 (y-axis) and positive marker F1 (x-axis); and (e) scatter plot between purity F1 (y-axis) and total transcripts per cell (x-axis). All the loss functions except the cell-calling loss increased the expression purity. The cell-calling loss increased the number of transcripts captured, without detriment to expression purity. The nucleus encapsulation loss improved purity metrics, by leveraging genes expressed in nuclei to guide the capture of genes expressed in cells. The overlap and over-segmentation losses improve the shapes of predicted cells, which translates into improved purity metrics. The differences between `BIDCell` and `loss_posneg` show that the marker losses improved performance considerably, particularly in purity metrics and Pearson correlation to Chromium data, while more transcripts were captured in the segmented cells.

Supplementary Figure 5: Illustrations of the segmentation predictions from different ablation experiments, where each loss was individually set to zero as indicated by `loss_`. The segmentations without cell-calling loss tend to be too small, while they are too large when marker losses are excluded. Exclusions of the other losses reduced the correspondence of the cell boundaries to the input images.

- Including the Cellpose dot in the ablation experiment in Supp. Fig. 3 panels c-e would help with comparing the methods.

Response: We thank the reviewer for this suggestion and we have included all other benchmarked methods (coloured in different shades of greys) in the supplementary figures related to the ablation experiments for comparison.

- Did the authors test other values besides $\lambda=1$ (other than the ablation experiment)? Would cross validation allow these lambdas to be tuned according to each data set and perhaps result in better performance?

Response: In our revised manuscript, we added results using different weights (λ values) besides 1.0 (Supplementary Figure 6). The λ of individual loss functions was set to either 0.5 or 2.0, while all other λ s were kept at 1. The results demonstrate that the performance of BIDCell may be further improved by tuning the λ s. Overall, Pearson correlation and purity metrics increase with higher weights for positive marker, nuclei, and overlap losses, and lower weights for the cell-calling loss. These findings are consistent with expectations. Additional results are shown in Supplementary Figure 6.

Supplementary Figure 6: Performance of BIDCell with different weights (*lambdas*) for each loss function. (a) Box plots of the number of transcripts and genes per cell for the different settings; (b) scatter plot showing genes expressed in the segmented cells compared to the nuclei; (c) scatter plot between Pearson correlation with Chromium (y-axis) and total transcripts per cell (x-axis); (d) scatter plot between inverse negative marker F1 (y-axis) and positive marker F1 (x-axis); and (e) scatter plot between purity F1 (y-axis) and total transcripts per cell (x-axis). BIDCell performs well across various datasets and technologies when using default weights for all losses (set to 1.0), and metrics may be further improved by setting different weights. Pearson correlation and purity metrics were higher with higher weights for positive marker, nuclei, and overlap losses, and lower weights for the cell-calling loss.

The loss functions of BIDCell were designed such that tuning of the *lambdas* is not necessary. Using the default 1.0 for all lambda values, we have shown that BIDCell performs well, and it outperformed existing methods across various datasets and technologies. The user may choose to tune the *lambdas* to enhance segmentation results (e.g., with cross-validation), but this will take more time. Hence, we believe it is up to each user to decide whether the slight enhancement of segmentation results warrants the additional computational time.

Minor points

1) It seems to me that for cell segmentation, people care mostly about a) the cell typing/purity of transcript assignments and b) the visual look of the segmentation. I'm not sure that the rest of the metrics introduced in CellSPA are particularly useful. In particular, the authors should at least mention what conclusions (if any) we can draw from looking at the panels of Figure 2. For example, number of transcripts per cell - is more necessarily better? Couldn't some cell types have less transcripts per cell?

Response: We thank the reviewer for this comment. As described in lines 162-170, these metrics can be grouped into five distinct yet complementary categories, each highlighting different properties of the segmented cells. In particular, the metrics shown in Figure 2 are mainly the metrics in the first category, which are the baseline (QC) characteristics that primarily characterise the segmented cells and determine the minimal appropriateness of a cell segmentation method, against a certain threshold. For example, for the number of transcripts per cell, we anticipate the segmented cells to exhibit values similar or higher than the Chromium data (represented by the grey-filled box).

However, a higher value is not always indicative of better performance as it can also suggest overgrowth of the cell boundary, where we discuss in more details in the later section (Figure 3). This also applies to % of cells expressing each gene. Therefore, the focus of this section is to determine whether a method offers additional information to the nuclei. Methods meeting this standard will be examined further in the next section using metrics in Categories 2-5.

Should the phrase “% of cells expressed per gene” be “% of cells expressing each gene”?

Response: Thank you and rephrased as suggested.

2) The ablation study is referred to in Results Section 1, Supplementary Figure 3, but not explicitly mentioned as “ablation study” in the main text until line 244 (“As an additional sensitivity analysis to the ablation study”)

Response: Thank you and we have revised Results Section 1 as:

“We investigated removing individual losses in an ablation study with Xenium-BreastCancer1 data (Supplementary Figures 4 and 5).”

3) Figure 1 - In “Deep learning model” - the abbreviations E1-E5, D1-D4 should be explained in the text or figure legend, in addition to the dotted lines and the meaning of the colors of the dotted lines.

Response: As suggested, we have added explanations to the abbreviations and the colours of the lines connecting different layers in the caption of Figure 1.

“In the deep learning model, E1 to E5 and D1 to D4 are respectively the encoding and decoding layers, while the connectivity between layers to each decoding layer is indicated by arrows of a unique colour (e.g., green for D3).”

4) Figure 2 - should say that ‘p’ in the elongation figure is correlation or change it to r or cor in line with other areas of the text, otherwise it’s easy to confuse with p-value.

Response: Thank you and we changed the “p” to “cor” as suggested.

Methods

- Apologies if I missed this somewhere, but is it “self-supervised” because you use a separate procedure to generate labels prior to fitting the deep learning model?

Response: We thank the reviewer for the question. Please see our extended response on page 20 for comment #3.1. Please note that we have revised our manuscript (BIDCell Training and Loss Functions section in the Materials and Methods) to better describe our framework.

- How are overlapping patches aggregated if there is disagreement in the overlapping regions?

Response: We simply use different regions from the overlapping patches as illustrated in the new Supplementary Figure 25, without additional operations for disagreement. This is done to minimise boundary effects such as sharp or cut-off cells in the final prediction. We have also added more explanation to Practical Implementation in Materials and Methods.

Supplementary Figure 25: Illustration of the aggregation of patches to minimise border effects. When dividing the transcriptomic maps into patches, we create two sets of patches of the same lateral dimensions with an overlap equal to half the lateral size of the patches. The first set (green borders) is obtained from a straightforward grid partitioning, while the second set (yellow border) overlaps the first set. The final prediction comprises the purple regions from the first set, and the blue regions from the second set.

“The patch-based predictions could result in effects along the patch boundaries such as sharp or cut-off cells. When dividing the transcriptomic maps into patches, we create two sets of patches of the same lateral dimensions with an overlap equal to half the lateral size of the patches. The predictions for the patches were combined (see Supplementary Figure 25), without additional operations to resolve potential disagreement between predictions of the two sets. Only patches from the first set (no overlaps) were selected during training, while all patches were used during inference.”

Data and source code

It seems the data and source code is available on the public BIDCell GitHub repository. Currently, the authors' code is in the form of a collection of standalone python scripts that are run in succession to preprocess, train, and postprocess. I think this is OK but ideally could be converted into a package form rather than sequentially running python scripts. In its current form, it would probably take some time to edit these steps for another data set, but this is probably unavoidable given the complex nature of the segmentation problem.

Response: We thank the reviewer for the suggestions. We have developed a Python package for BIDCell and its associated functions, including data preprocessing. The code may be found at <https://github.com/SydneyBioX/BIDCell>. We have added new functionalities to generalise to the data formats of different platforms, e.g., Xenium, CosMx, MERSCOPE, and Stereoseq. We include instructions in the README, and also examples for the entire data processing and segmentation pipeline for different platforms to make the code easier to use. Furthermore, we plan to continue supporting and extending the code after this manuscript.

In addition, following the instructions as written on the main GitHub README, I ran the steps without changing anything and encountered the error “IndexError: Target -9223372036854775808 is out of bounds” when running the python train.py --config_file configs/config.json. Not sure if the authors have seen this before.

Response: We thank the reviewer for raising this issue. We have found the cause of this error to be from a bug with the code for the positive and negative marker losses. The bug caused this value -9223372036854775808 (-2^{63}) to be generated, which is an invalid prediction class target. We have fixed this bug and pushed it to our GitHub repository.

Reviewer #3 & 4 (Remarks to the Author):

The authors propose a novel methodology to refine image-based spatial transcriptomics cell segmentation called biologically-informed deep learning-based cell segmentation (BIDCell) that takes into account the molecular data. They also present an evaluation suit: Cell Segmentation Performance Assessment (CellSPA) which compares performance measured with a variety of metrics. They compare different methods extensively using publicly available data acquired with different technologies.

The paper provides extremely interesting metrics for the very difficult problem of evaluating segmentation when ground-truth data is unavailable, for example, evaluating the segmentation quality given positive and negative cell marker gene expression, comparing the tradeoff between transcripts per cell, and correlation with scRNA-seq results. These metrics are implemented on the companion software CellSPA.

In our opinion, a major point to improve is the following: we had difficulty appreciating the value of some of the metrics. We suggest that the proposed evaluation benchmark should be improved by providing more extensive details of how each metric benefits the estimation of the similarity between the segments and the true unknown correct cell segments.

The remaining comments are itemized below so they can be objectively answered and quoted.

Comments:

1. Why is the method “self-supervised” if it uses negative and positive markers on the training loss? Wouldn't “weakly-supervised” be more appropriate?

Response: Recently, deep learning methods that do not require the need for complete annotation of the dataset have been popular. These methods typically leverage ideas from paradigms including unsupervised, weakly supervised, and self-supervised learning (SSL). Some methods combine ideas from more than one paradigm. A core aspect of SSL that we adopt is that it automatically computes supervisory signals by generating labels from the input data (e.g., specific properties or parts of data) to train the model (Xie et al. 2023).

We agree that the positive and negative marker losses act as weak supervision, since they provide partial targets for cell segmentations. Labels tend to be manually annotated for weakly supervised methods however, which is in contrast to SSL, where labels are derived automatically from the input data (or obtained from a related dataset) (Hung et al. 2019). BIDCell uses a single-cell reference as a third input modality, which is used to generate the labels for the marker losses. This is an automated process (there is no manual input) as the markers are the genes with the top or bottom 10 percentile of expressions. We found that a simple cutoff was sufficient across various datasets.

We have revised our manuscript (BIDCell Training and Loss Functions section in the Materials and Methods) to better describe our framework:

“The BIDCell framework combines several loss functions that automatically derive supervisory signals from the input data and/or predicted segmentations at each step of the training process. This approach to learning is a core aspect of SSL (Xie et al., 2022). Furthermore, the modular and additive design of the loss functions allows each loss to be swapped out with alternative approaches to compute training signals. The SSL label describes the ability of the framework to automatically learn relationships between gene expressions and cell morphology from its inputs.”

The nuclei encapsulation, cell-calling, over-segmentation, and overlap losses guide the basic morphology of cells. The positive and negative marker losses refine the cell morphologies learned through the other loss functions, by further guiding the model to learn biologically-informed relationships between gene expressions and cell morphology. This is reminiscent of the pretext and downstream (fine-tuning) stages commonly encountered in SSL, where the pretext task aids the model to learn better representations or intermediate weights, while the fine-tuning task refines the weights and further improves performance for a particular prediction task.”

Refs:

Hung, Wei-Chih, Varun Jampani, Sifei Liu, Pavlo Molchanov, Ming-Hsuan Yang, and Jan Kautz. 2019. “SCOPS: Self-Supervised Co-Part Segmentation.” In *2019 IEEE/CVF Conference on Computer Vision and Pattern Recognition (CVPR)*. IEEE. <https://doi.org/10.1109/cvpr.2019.00096>.

Xie, Yaochen, Zhao Xu, Jingtun Zhang, Zhengyang Wang, and Shuiwang Ji. 2023. “Self-Supervised Learning of Graph Neural Networks: A Unified Review.” *IEEE Transactions on Pattern Analysis and Machine Intelligence* 45 (2): 2412–29.

2. Segments should be overlaid over nuclei and cytoplasm (or membrane) channels, H&E could be used if that’s not available because it’s harder to evaluate the segmentation masks over transcripts.

Response: We have added figures where the segmentation is overlaid over DAPI images in the revised manuscript. Cytoplasm, membrane, and H&E images are not available for all datasets (e.g., H&E is only available for Xenium-BreastCancer1), and hence may not be used in a consistent presentation of results. We believe that an important aspect of segmentation methods is their ability to capture transcripts within cells, and the transcript-overlaid figures give the reader a visual indication of this ability. Hence, we have included the additional figures in Supplementary Figure 3:

a Xenium-BreastCancer1

b Xenium-MouseBrain

Supplementary Figure 3: Illustration of *BIDCell* and other segmentation methods overlaid on DAPI images for (a) *Xenium-BreastCancer1* and (b) *Xenium-MouseBrain*.

3. Could you elaborate on how some points are below the red diagonal line in Supplementary Figure 4? From my understanding, a “segmented cell” is a superset of a “nucleus”. Therefore, it should always contain more transcripts. Is it because these methods do not take the initial “nucleus” segmentation as a hard constraint?

Response: We thank the reviewer for the attention to detail. We agree that cells should contain more transcripts than nuclei. We investigated the few cells where this was not the case, and found that it was caused by an issue with the morphological processing code. For these cells, the code was not fully considering their connectivity, meaning sections of cells were sometimes missing. We have fixed the code to eliminate such occurrences, and all the cells are now above the red diagonal line in the figure (see the new Supplementary Figure 9).

We note that for the Voronoi method, it is possible for cells to be below the red diagonal line. The seeds were single pixels (centroids of nuclei), and depending on the shape of the predicted tessellation, the cell’s border might cut through (and not fully overlap) a nucleus that was predicted by Cellpose (which informs the x-axis).

4. Data from CellSPA data should be stored in permanent and versioned storage and not google drive. Is there a difference with the data of the 10x resource?

Response: We thank the reviewer for this comment. We have updated our CellSPA package and the example data now is saved within the package. The users can run the example without downloading the data separately.

5. The authors could spend more time explaining how each metric contributes to better evaluating the segmentation performance. Otherwise, it's not clear how each individual metric benefits the CellSPA package.

Response: We thank the reviewer for this comment. Lines 162-170 describes that all individual metrics can be broadly classified into five complementary groups that capture different properties of the segmented cells, contributing in evaluating the cell segmentation from different aspects. Within each group, the individual metrics are listed in Table 3, where we describe in detail how each metric is calculated and what level these metrics are examining.

6. Material & methods L186: Why are the background probabilities averaged? Won't this overestimate the background since each cell's instance probability prediction would classify other cells' instances as background?

Response: For an input image patch containing n cells, the input tensor to the model is permuted such that there are n patches of predicted background probabilities. We average the background probabilities to obtain a common set of pixels for background that is consistent across all cells, to avoid potential disagreement between background pixels predicted for different cells. We have demonstrated that this approach does not result in the overestimation of background (i.e., cells that are too small), as the cells capture a reasonable amount of transcripts, as shown in Figure 2, and that BIDCell outperforms other methods across various metrics.

7. Which lambda (without subscript) is referred to in L289 of material & methods? Is it all of them? It could be more clear.

Response: We apologise for the confusion. This lambda is referring to all the loss functions. We have revised the Materials and Methods on page 10:

"The value of λ for all loss functions was set to 1.0".

8. It has been shown that torch Adam weigh-decay is not L2 regularization [1]. Therefore, equation 13 from material & methods should be corrected.

Response: We thank the reviewer for raising this point. We agree that weight decay and L2 regularisation are not equivalent for the Adam optimiser. Despite the PyTorch documentation stating that weight decay is the L2 penalty, and the source code showing that the weight decay component is added to the gradients (suggesting L2 regularisation), there is some confusion within the community about its exact implementation. Hence, to avoid confusion in our manuscript, we have removed the L2 regularisation term in Equation 13, and made a note that we used a weight decay of 0.0001 where we detail the other hyperparameters of the Adam optimiser under the section Practical Implementation.

9. Why was the watershed baseline (L363 from material & methods) run on the sum of the molecule channels and not the cytoplasm or membrane channel? The sum of molecule channels does not yield a topology that is adequate for the watershed operator, which splits segments on the high-gradient features.

Response: We agree that the Watershed algorithm may be applied to the cytoplasm or membrane channel. However, the availability of such types of images is limited by the current state of SST

technologies. All the methods compared in our study used only DAPI and/or transcriptomic images, and we used the same inputs for the Watershed algorithm as the other methods for fair comparison.

10. Why are Circularity (L395 from material & methods) and Compactness (L406) needed when they are extremely similar?

Response: We thank the reviewer for their comment. While we agree that both measures are similar, there is a key difference that may offer a different metric in our evaluation. The main difference is that Circularity uses the convex perimeter and is thus insensitive to local irregularities, whereas Compactness uses the perimeter of the cell, thus considering local irregularities in the cell perimeter. We have revised their definitions in the Cell Morphology Metrics section of Materials and Methods to better describe their differences.

“Circularity measures the area to perimeter ratio while excluding local irregularities of the cell. We used the convex perimeter of the object as opposed to its true perimeter to avoid concave irregularities. The value will be 1 for a circle and decreases as a cell becomes less circular.

Compactness measures the ratio of the area of an object to the area of a circle with the same perimeter. Compactness uses the perimeter of the cell thus it considers local irregularities in the cell perimeter. A circle will have a value of 1, and the less smooth or more irregular the perimeter of a cell, the smaller the value will be. For most cells the numerical values for compactness and circularity are expected to be similar. Identifying which cells have large differences between these metrics can identify cells with highly irregular perimeters which may be of interest for downstream analysis and quality control for segmentation.”

11. How much does the morphological post-processing (L192 from material & methods) impact the proposed method results? Several metrics evaluate the convexity and smoothness of the contours in some way, which are improved by the morphological post-processing. Wouldn't this post-processing also benefit the other methods?

Response: We apply morphological operations (dilation and erosion with 5x5 kernels, and removing small isolated islands) to the initial segmentation predictions of the CNN. These are standard operations that close holes and fill gaps between pixels of a mask. This is necessary to ensure that the pixels of a cell are connected together, as there is no hard constraint about pixel connectivity in the CNN. After morphological processing, the number of transcripts captured within each cell is slightly higher, while purity metrics and correlation with Chromium are the same or slightly higher. The morphological operations have no effect where the pixels are already connected together for a cell. The only discernible difference for the outputs of the other methods are where there are holes (JSTA and a few cells from Baysor), otherwise, there is no effect on the other methods. There would be no effect on metrics such as convexity and smoothness, since these are computed using the cell perimeter.

We have added results of predicted segmentations before and after morphological operations in the revised manuscript in Supplementary Figure 24 and Figure R5.

Supplementary Figure 24: Performance of BIDCell with or without morphological processing. (a) Example images to illustrate BIDCell before and after morphological processing; (b) scatter plot between Pearson correlation with Chromium (y-axis) and total transcripts per cell (x-axis); (c) scatter plot between inverse negative marker F1 (y-axis) and positive marker F1 (x-axis); and (d) scatter plot between purity F1 (y-axis) and total transcripts per cell (x-axis).

Figure R5: Example segmentations predicted by other methods, before and after morphological processing. Here, we applied the same operations as BIDCell.

12. Why is the entropy compared with the coefficient of variation (CV) using Pearson correlation? Is there any reason why this relationship should be linear? Would a rank-based correlation be more appropriate since the CV depends on the scale of the distribution?

Response: We thank the reviewers for this suggestion. We use Pearson correlation based on the observation that most of these relationships in the real data display a close to linear relationship. For Figure 3g, a Spearman based correlation produces similar results: Pearson (0.42, 0.44, 0.28) vs Spearman (0.47, 0.47, 0.28).

13. Could the authors provide more details on the Neighbouring Contamination metric? A figure would really assist in understanding it.

Response: We have now included the schematic figure in Supplementary Figure 8.

Supplementary Figure 8: Schematic figure for neighbouring contamination. A pair of cell type A and B and cell type A's marker a is predefined. For each cell type B, the distance to the nearest cell type A is calculated. The cell of cell type B is then grouped based on their distance to the nearest cell of cell type A (4 groups in the figure). The proportion of cells expressed marker a is then calculated.

Minor comments:

1. Use the same ggplot2 theme as the rest of the article in Supplementary Figure 1.

Response: We have changed the theme of the Supplementary Figure 1 to be consistent with the rest of the figures in the article.

2. Supplementary Figure 3 could be improved by switching from sequential to high-contrast colors.

Response: Thank you and we have updated our Supplementary Figure 3 with high-contrast colours.

3. It would greatly improve the article to have the plots represented as vector graphics, especially because several plots need to be zoomed in. For example, Figures 1,2,3, etc.

Response: We thank the reviewers for this comment. These figures were originally in vector graphics format but were compressed when inserted in the word documents. We have now also uploaded the vector graphics in the resubmission.

4. Should it be “ $p = \dots$ ”, “ $R = \dots$ ” or “ $\text{cor} = \dots$ ” on Figure 2e? It would be helpful to provide more information on what this measurement means, besides correlation, because there are several methods of correlation.

Response: Thank you and we have changed the “ $p=$ ” to “ $\text{cor}= \dots ” and noted this is the Pearson correlation in our caption.$

5. Typo on figures' legend, Firoblast -> Fibroblast

Response: Thank you, we have fixed this typo in the related figures.

6. Figure 3a correlation should be normalized between 0 and 1.0 to avoid misleading an unattentive reader.

Response: We have updated the scale of the colour in Figure 3a as suggested.

7. L338 material & methods, Table 1 should be Table 2.

Response: We apologise for this and have corrected this to Table 2, and have checked all other cross-references in the manuscript.

8. L431 material & methods, Table 2 should be Table 1.

Response: We apologise for this and have corrected this to Table 1, and have checked all other cross-references in the manuscript.

9. Shouldn't the Supplementary Figure 8 axis be on the same scale between different marker genes? If not, why not?

Response: We thank the reviewer for this suggestion. We find that the range of these evaluation metrics are cell type dependent and using different scales in this case can better illustrate the relative rankings of the method.

10. How significant is the difference between 1 - Neg F1 and Pos F1 scores from Supplementary Figures 12d and 13d? They are all on the second decimal level.

Response: We thank the reviewer for this comment. We note that the 1 - Neg F1 and Pos F1 scores from the original Supplementary Figures 12d and 13d are the average values of the whole dataset. To better illustrate the difference between different methods, we add two additional figures (Supplementary Figures 18 and 20) which illustrate 1 - Neg F1 vs Pos F1 by cell type.

Supplementary Figure 18: Expression purity benchmarking results for CosMx-Lung: scatter plots showing the positive marker F1 scores vs. the 1 - negative marker F1 scores for each of the cell type, where each dot is one method.

Supplementary Figure 20: Expression purity benchmarking results for MERSCOPE- Melanoma: scatter plots showing the positive marker F1 scores vs. the 1 - negative marker F1 scores for each of the cell type, where each dot is one method.

Reviewer #1 (Remarks to the Author):

The majority of the comments have been satisfactorily addressed by the authors, however certain findings are still troubling. I think this would make a great article if it were properly described. First, I will provide a fair and impartial assessment of this work, focusing on the benchmark testing, algorithm innovation, results section, and code usability. I don't have any animosity toward this piece, only anticipation. I'm only giving the article a logical and impartial appraisal. If it hampered your revision in any way, I apologize.

1) My comment of the lack of a thorough performance comparison is legitimate. The author really needs to compare with all published algorithms (rather than selecting representative algorithms for comparison as you said). If it doesn't apply please explain (that's what the first comment stated). This is not my intention to make things difficult for you. In addition, quantitative indicators allow readers to interpret your article more clearly, which is also a must.

2) The algorithm's usability and scalability will be significantly enhanced by the use of this approach in BGI Stereo-seq. The paper also mentions this technique, and it makes sense to use it in BGI Stereo-seq. Most published articles provide raw data and are not, as the authors reply, "While a number of papers claim the public availability of BGI...". Although the Stereo-seq data set was used for testing by the authors, this does not appear to be done voluntarily. I believe stereo-seq data can also be used with methods like SCS. It appears that comparing StereoCell alone is insufficient to demonstrate the dependability of BIDCell. The author doesn't need to start with "That being said...".

3) Model assessment is imprecise and unreliable. According to some of the author's studies, cellpose appears to produce superior outcomes than BIDCell. Reviewer 2 also raised this aspect of the issue. The quantitative statistics indeed show that cellpose beats BIDCell. Your response is appropriate, but you just need to provide a thorough explanation; the author doesn't need to start with "That being said...".

Reviewer #2 (Remarks to the Author):

Overall, the authors addressed the comments well. I have two minor remaining comments.

1) The terminology of self-supervised learning is still unclear to me. I appreciated the explanation in the response to Reviewer 3 #1. However, it is confusing because even though determining marker genes via ranking is "automatic" from the scRNA-seq, this process is equivalent to providing the model with marker gene labels, which would make it more in line with weak supervision as reviewer 3 mentioned. I see that the authors added more explanation on this in the Methods, but it would be great to also add a sentence clarifying why BIDCell is SSL, either in the last paragraph of the introduction or in the first paragraph of the first results section.

2) There are still a couple subjective phrases in the Results section. Ideally, the Results should remain objective.

a) "Ideal balance" - This could be fixed by replacing the sentence "Figure 3a illustrates that BIDCell achieves the ideal balance between high correlation with segmented nuclei and a large cell body among all methods" with something like "Among all methods, BIDCell achieves the second highest Pearson correlation with Chromium expression ($r=xx$) while also detecting more transcripts per cell than Cellpose (Figure 3a)."

b) "Next, by examining... BIDCell achieves a clear improvement ..." - this sentence could be replaced with something like "BIDCell also has a higher presence of positive markers and a lower presence of negative markers in large cells (Figure 3d and Supplementary Figure 13), demonstrating an improvement in the expression purity of segmentation."

Reviewer #3 (Remarks to the Author):

I appreciate the authors' efforts in addressing all the concerns and questions raised in the first round of review.

The authors have addressed each point previously highlighted by us.

They also added results for new technology (BGI Stereo-seq), as requested by reviewer 1, which can take a considerable amount of work.

Reviewer #4 (Remarks to the Author):

The Authors have addressed our points satisfactorily.

We are still a bit unsure about the 'self-supervision' part, but the author's argument is sufficient, and we accept that the boundary between supervised and self-supervised is fuzzy sometimes. Other than that, we are satisfied with the revision -- the manuscript has improved.

REVIEWER COMMENTS

Reviewer #1 (Remarks to the Author):

The majority of the comments have been satisfactorily addressed by the authors, however certain findings are still troubling. I think this would make a great article if it were properly described. First, I will provide a fair and impartial assessment of this work, focusing on the benchmark testing, algorithm innovation, results section, and code usability. I don't have any animosity toward this piece, only anticipation. I'm only giving the article a logical and impartial appraisal. If it hampered your revision in any way, I apologize.

1) My comment of the lack of a thorough performance comparison is legitimate. The author really needs to compare with all published algorithms (rather than selecting representative algorithms for comparison as you said). If it doesn't apply please explain (that's what the first comment stated). This is not my intention to make things difficult for you. In addition, quantitative indicators allow readers to interpret your article more clearly, which is also a must.

Response: We appreciate the comments. As alluded to by the reviewer, not all methods are applicable. We have added explanations in Supplementary Table 1, and added reasons to the Performance Evaluation section in Materials and Methods. Due to the various limitations of the methods, their inclusion would cast them in a negative light, and were thus omitted. We agree that quantitative indicators are important, which was the motivation behind our CellSPA framework, and our inclusion of multiple metrics in the Results and Supplementary.

Types	Name of method	Reference	Included	Exclusion reason
Adapted from classical approach	Dilation of nuclei		Y	
	Voronoi		Y	
	Watershed		Y	
Deep learning-based	Cellpose	Stringer et al., 2021	Y	
	JSTA	Littman et al., 2020	Y	
	GeneSegNet	Wang et al., 2022	N	Unclear how data should be processed into the required format and used to generate initial labels, missing instructions and/or code
	SCS	Chen et al., 2023	N	No cells were detected for Xenium, CosMx, and MERSCOPE, and considerably lengthy runtimes compared to other methods (above 72-hour practical requirement) for Stereo-seq
	Mesmer	Greenwald et al., 2022	N	Poor performance (neighbouring contamination); consistently rounded morphologies that do not demonstrate the diverse morphologies expected for the cell types
Transcript-based	Baysor	Petukhov et al., 2022	Y	
	pCiSeq	Qian et al., 2020	N	Method designed to assign transcripts to cells, does not consider cell boundaries and their overlap
	ClusterMap	He et al., 2021	N	Poor performance (cells too large and few in number)
	Sparcle	Prabhakaran, 2022	N	Method designed to assign transcripts to cells, does not consider cell boundaries and their overlap
	StereoCell	Li et al., 2023	N	Poor performance (neighbouring contamination); consistently rounded morphologies that do not demonstrate the diverse morphologies expected for the cell types

“Methods that were excluded from the evaluations include those that focus on the assignment of transcripts to cells and do not consider the cell boundaries, underperformance on the public datasets, lack of code and instructions to prepare data into the required formats, and failure of the method to detect any cells (Supplementary Table 1).”

2) The algorithm's usability and scalability will be significantly enhanced by the use of this approach in BGI Stereo-seq. The paper also mentions this technique, and it makes sense to use it in BGI Stereo-seq. Most published articles provide raw data and are not, as the authors reply, "While a number of papers claim the public availability of BGI...". Although the Stereo-seq data set was used for testing by the authors, this does not appear to be done voluntarily. I believe stereo-seq data can also be used with methods like SCS. It appears that comparing StereoCell alone is insufficient to demonstrate the dependability of BIDCell. The author doesn't need to start with "That being said...".

Response: We appreciate the comments. We agree that applying BIDCell on Stereo-seq data further demonstrates its flexibility, and have included this in the revised manuscript. Regarding data availability, we were unable to access some published raw Stereo-seq data due to submission complexity with the China National Genebank Database (CNCDB).

We attempted to run SCS on the Stereo-seq embryo data on two different servers. We were able to run the code on this dataset after manually fixing some issues, such as very high CPU usage and missing packages, and by reaching out to the authors to resolve crashes. However, we found SCS to have a considerably lengthy runtime compared to other methods. On our servers (Titan V or A5000 GPUs), the runtime (after a 2 days pilot run) is estimated to be over 10 days. This is above the 72-hour practical requirement we used for the other methods. Though the number of epochs could be reduced from the default setting of 100, this may not provide a good representation of the method. Here, we exclude SCS for this dataset and state the exclusion reason in Supplementary Table 1.

We have added results for Cellpose applied to this Stereo-seq embryo data.

We have rephrased our initial response:

We believe that our ability to extend to three technologies will enable the flexibility to extend to a fourth. We have strengthened the manuscript by following the reviewer's suggestions to apply BIDCell to BGI Stereo-seq data. We select Stereo-seq data that can be publicly downloaded from a website and here, we apply BIDCell to the Stereo-seq mouse embryo dataset from Chen and colleagues published in Cell. (Chen et al. 2022). We have added the results to Supplementary Figure 21 and added details about this dataset and results in our manuscript:

Ref: Chen, Ao, Sha Liao, Mengnan Cheng, Kailong Ma, Liang Wu, Yiwei Lai, Xiaojie Qiu, et al. 2022. "Spatiotemporal Transcriptomic Atlas of Mouse Organogenesis Using DNA Nanoball-Patterned Arrays." *Cell* 185 (10): 1777–92.e21.

Under Results on MS page 6:

"Furthermore, we also applied BIDCell to Stereo-seq from BGI (Supplementary Figure 21). We have now demonstrated the applicability of BIDCell on data from four major platforms, and from five different tissue types. We believe that our method has the flexibility and generalisability to other data from other SST platforms and tissues."

Under Materials and Methods on pages 2 and 3:

Stereo-seq-MouseEmbryo. The Stereo-seq data used in this study, including the DAPI image and detected gene expressions (bin 1), were downloaded from <https://db.cngb.org/stomics/mosta/download/> for sample E12.5_E1S3. Stereo-seq data contains a

far greater number of genes compared to Xenium, CosMx, and MERSCOPE. For efficiency, we selected a panel of 275 highly variable genes (HVGs) as the input to BIDCell. The HVGs are the common genes of the top 1,000 HVGs from both Stereo-seq data and the single-cell reference data.

Mouse Embryo reference. The reference for Stereo-seq-MouseEmbryo was downloaded from GEO database under accession code GSE119945: <https://www.ncbi.nlm.nih.gov/geo/query/acc.cgi?acc=GSE119945> (Cao et al., 2019), which contain both counts and cell type annotation data. The E12.5 data was then used as reference.

Supplementary Figure 21: Application of BIDCell to mouse embryo data from BGI Stereo-seq. (a) Example cell segmentations from BIDCell, StereoCell, and Cellpose; and (b) examples of marker expression of spatial regions for Chen et al. (Bin 50), BIDCell, StereoCell, and Cellpose.

3) Model assessment is imprecise and unreliable. According to some of the author's studies, cellpose appears to produce superior outcomes than BIDCell. Reviewer 2 also raised this aspect of the issue. The quantitative statistics indeed show that cellpose beats BIDCell. Your response is appropriate, but you just need to provide a thorough explanation; the author doesn't need to start with "That being said...".

Response: We would like to clarify that our model assessment framework provides multiple aspects of comparison for cell segmentation. In the absence of a ground truth in cell segmentation tasks, it is vital to consider multiple aspects at the same time when comparing different methods. Taking all aspects in our framework into account, Cellpose does not perform better than BIDCell.

We agree that a thorough explanation is needed, and thus, following the suggestions of Reviewer 2, we have reworded some of the sentences in the Results section to objectively describe our method performance. Furthermore, please note that we included a detailed discussion in our discussion section (page 8, lines 341 to 352) to thoroughly explain and discuss the importance of a variety of segmentation metrics in systematic benchmarking of the cell segmentation methods.

Reviewer #2 (Remarks to the Author):

Overall, the authors addressed the comments well. I have two minor remaining comments.

1) The terminology of self-supervised learning is still unclear to me. I appreciated the explanation in the response to Reviewer 3 #1. However, it is confusing because even though determining marker genes via ranking is "automatic" from the scRNA-seq, this process is equivalent to providing the model with marker gene labels, which would make it more in line with weak supervision as reviewer 3 mentioned. I see that the authors added more explanation on this in the Methods, but it would be great to also add a sentence clarifying why BIDCell is SSL, either in the last paragraph of the introduction or in the first paragraph of the first results section.

Response: We thank the reviewer for the suggestions. We have added the following sentence to the first paragraph of Results:

"The BIDCell framework automatically derives supervisory signals from the input data and/or predicted segmentations, which is an approach to learning that we borrow from SSL."

2) There are still a couple subjective phrases in the Results section. Ideally, the Results should remain objective.

a) "Ideal balance" - This could be fixed by replacing the sentence "Figure 3a illustrates that BIDCell achieves the ideal balance between high correlation with segmented nuclei and a large cell body among all methods" with something like "Among all methods, BIDCell achieves the second highest Pearson correlation with Chromium expression ($r=xx$) while also detecting more transcripts per cell than Cellpose (Figure 3a)."

Response: We thank the reviewer for the suggestions. We have reworded this sentence as follows to improve objectivity and emphasise the importance of employing a variety of metrics to quantify segmentation performance:

"Figure 3a demonstrates the importance of employing two metrics to quantify segmentation performance. While Cellpose achieved the highest Pearson correlation overall, BIDCell achieved the highest Pearson correlation among methods that detect a similar number of transcripts as Chromium data (i.e., cell sizes that are more similar to segmentation of the cell body as opposed to the nuclei)."

b) "Next, by examining... BIDCell achieves a clear improvement ..." - this sentence could be replaced with something like "BIDCell also has a higher presence of positive markers and a lower presence of negative markers in large cells (Figure 3d and Supplementary Figure 13), demonstrating an improvement in the expression purity of segmentation."

Response: We have reworded the sentence as suggested.

Reviewer #3 (Remarks to the Author):

I appreciate the authors' efforts in addressing all the concerns and questions raised in the first round of review.

The authors have addressed each point previously highlighted by us.

They also added results for new technology (BGI Stereo-seq), as requested by reviewer 1, which can take a considerable amount of work.

Response: We thank the reviewer for the positive comments.

Reviewer #4 (Remarks to the Author):

The Authors have addressed our points satisfactorily.

We are still a bit unsure about the 'self-supervision' part, but the author's argument is sufficient, and we accept that the boundary between supervised and self-supervised is fuzzy sometimes.

Other than that, we are satisfied with the revision -- the manuscript has improved.

Response: We thank the reviewer for the positive comments.

Reviewer #1 (Remarks to the Author):

I thank the authors for their efforts to address all concerns and issues raised in this second round of review.

I did find that SCS took a long time to run and was very hardware demanding. Other than that, we are satisfied with the revision -- the manuscript has improved.

REVIEWER COMMENTS

Reviewer #1 (Remarks to the Author):

I thank the authors for their efforts to address all concerns and issues raised in this second round of review.

I did find that SCS took a long time to run and was very hardware demanding. Other than that, we are satisfied with the revision -- the manuscript has improved.

Response: We thank the reviewer for the positive comments.